# SignAvatars: A Large-scale 3D Sign Language Holistic Motion Dataset and Benchmark

## Abstract

In this paper, we present SignAvatars, the first large-scale multi-prompt 3D sign language (SL) motion dataset designed to bridge the communication gap for Deaf and hard-of-hearing individuals. While there has been an exponentially growing number of research regarding digital communication, the majority of existing communication technologies primarily cater to spoken or written languages, instead of SL, the essential communication method for Deaf community. Existing SL datasets, dictionaries, and sign language production (SLP) methods are typically limited to 2D as the annotating 3D models and avatars for SL is usually an entirely manual and labor-intensive process conducted by SL experts, often resulting in unnatural avatars. In response to these challenges, we compile and curate the SignAvatars dataset, which comprises 70,000 videos from 153 signers, totaling 8.34 million frames, covering both isolated signs and continuous, co-articulated signs, with multiple prompts including HamNoSys, spoken language, and words. To yield 3D holistic annotations, including meshes and biomechanically-valid poses of body, hands, and face, as well as 2D and 3D keypoints, we introduce an automated annotation pipeline operating on our large corpus of SL videos. SignAvatars facilitates various tasks such as 3D sign language recognition (SLR) and the novel 3D SL production (SLP) from diverse inputs like text scripts, individual words, and HamNoSys notation. Hence, to evaluate the potential of SignAvatars, we further propose a unified benchmark of 3D SL holistic motion production. We believe that this work is a significant step forward towards bringing the digital world to the Deaf community. Our project page is at https://anonymoususer4ai.github.io/

## 1 Introduction

According to the World Health Organization, there are 466 million Deaf and hard-of-hearing people (Davis & Hoffman, 2019). Among them, there are over 70 million who communicate via sign languages (SLs) resulting in more than 300 different SLs across different Deaf community (The World Federation). While the field of (spoken) natural language processing (NLP) and language assisted computer vision (CV) are well explored, this is not the case for the alternate and important communicative tool of SL, and accurate generative models of holistic 3D avatars as well as dictionaries are highly desired for efficient learning (Naert et al., 2020).

We argue that the lack of large scale, targeted SL datasets is an important reasons for this gap putting a barrier in front of downstream tasks such as digital simultaneous SL translators. On one hand, existing SL datasets and dictionaries (Duarte et al., 2021; Albanie et al., 2021; 2020; Camgoz et al., 2018; Hanke et al., 2020; Huang et al., 2018) are typically limited to 2D videos or 2D keypoints annotations, which are insufficient for learners (Lee et al., 2023) as different signs could appear to be the same in 2D domain due to *depth ambiguity*. On the other hand, while parametric holistic models exist for human bodies (Pavlakos et al., 2019) or bodies & faces (Yi et al., 2023), there is no unified, large-scale, multi-prompt 3D holistic motion dataset with accurate hand mesh annotations, which are crucial for SL. The reason for this is that the creation of 3D avatar annotation for SL is a labor-intensive, entirely manual process conducted by SL experts and the results are often unnatural (Aliwy & Ahmed, 2021).

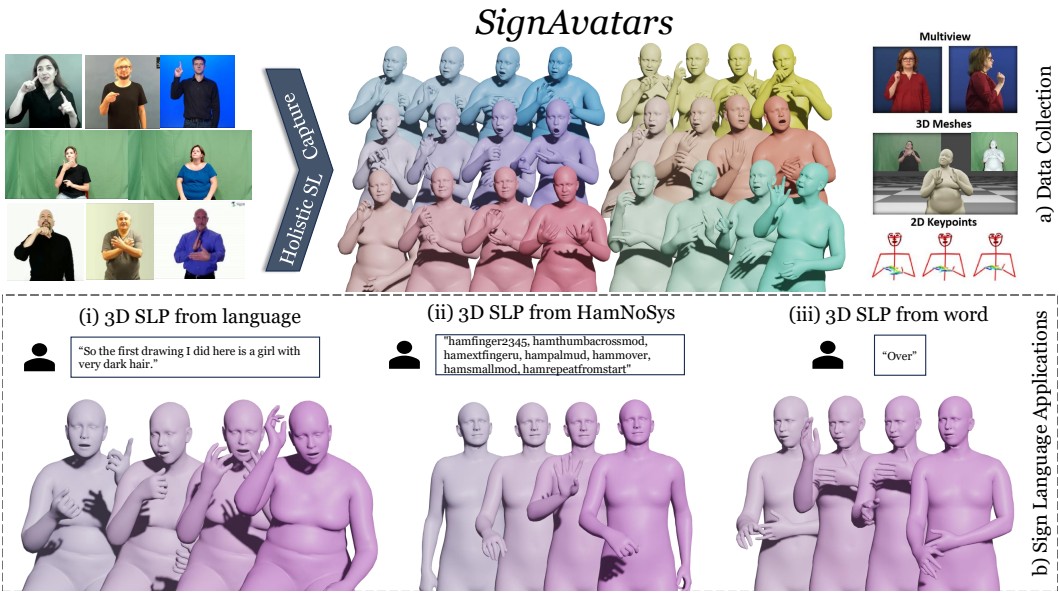

Figure 1: Overview of SignAvatars, the first public large-scale multi-prompt 3D sign language holistic motion dataset. (**upper row**) We introduce a generic method to automatically annotate a large corpus of video data. (**lower row**) We propose a 3D SLP benchmark to produce plausible 3D holistic mesh motion and provide a neural architecture as well as baselines tailored for this novel task.

To address this challenge, in this paper, we first introduce the SignAvatars dataset, gathering various data sources from public datasets to continuous online videos with mixed-prompt annotations including HamNoSys, spoken language, and word. Overall, we compile $70K$ videos from $153$ signers amounting to $8.34M$ frames. Unlike (Forte et al., 2023), our dataset is not limited to isolated signs, *i.e.* single sign per video, where HamNoSys-annotations are present, but includes continuous and co-articulated signs. To augment our dataset with 3D full-body annotations, including 3D body, hand and face meshes as well as 2D & 3D keypoints, we design an automated and generic annotation pipeline, in which we perform a multi-objective optimization over 3D poses and shapes of face, hands and body. Our optimizer considers the temporal information of the motion and respects the biomechanical constraints in order to produce accurate hand poses, even in presence of complex, interacting hand gestures. Apart from meshes and SMPL-X models, we also provide a *hand-only* subset with MANO annotations.

SignAvatars enables multitude of tasks such as 3D sign language recognition (SLR) or the novel 3D sign language production (SLP) from text scripts, individual words, and HamNoSys notation. To address this challenge and accommodate diverse forms of semantic input, we propose a novel approach utilizing a semantic Variational Autoencoder (VQVAE) (Van Den Oord et al., 2017) that effectively maps these varied inputs to discrete code indices. This *parallel linguistic feature generator* is fused with a discrete motion encoder within an auto-regressive model to generate sequences of code indices derived from these semantic representations, strengthening the text-motion correlation. Consequently, our method can efficiently generate sign motion from an extensive array of textual inputs, enhancing its versatility and adaptability to various forms of semantic information. We will demonstrate in Sec 5 that building such reliance and correlation between the low-level discrete representations leads to accurate, natural and sign-motion consistent SL production compared to direct regression from a high-level CLIP feature.

To quantitatively & qualitatively evaluate the potential of SignAvatars, we introduce a new benchmark and present the first results for 3D SL holistic mesh motion production from multiple prompts including HamNoSys, spoken language, and word. On this benchmark. We assess the performance of our Sign-VQVAE against the baselines we introduce, where we show a relative improvement of $200\%$. Though, none of these models can truly match the desired accuracy, confirming the timeliness and the importance of SignAvatars.

To summarize, our contributions are:

- We introduce SignAvatars, the first large-scale multi-prompt 3D holistic motion SL dataset, containing diverse forms of semantic input.
- To provide accurate annotations for SignAvatars, in the form of expressive 3D avatar meshes, we introduce a multi-objective optimization capable of dealing with the complex interacting hands scenarios, while respecting the biomechanical hand constraints. We initialize this fitting procedure by a novel multi-stage, hierarchical process.
- We provide a new 3D sign language production (SLP) benchmark for SignAvatars, considering multiple prompts and full-body meshes.
- We further develop a VQVAE-based strong 3D SLP network significantly outperforming the baselines, which are also introduced as part of our work.

We believe SignAvatars is a significant stepping stone towards bringing the 3D digital world and 3D SL applications to the Deaf community, by fostering future research in 3D SL understanding.

## 2 RELATED WORK

**3D holistic mesh reconstruction (for SL)**. Recovering holistic 3D human body avatars from RGB videos and parsing them into parametric forms like SMPL-X (Pavlakos et al., 2019) or Adam (Joo et al., 2018) is a well explored area (Yi et al., 2023; Pavlakos et al., 2019; Lin et al., 2023b). For example, Arctic (Fan et al., 2023) introduces a full-body dataset annotated by SMPL-X, for 3D object manipulation. Hasson et al. (2019) provide a hand-object constellations datasets with MANO annotations. However, such expressive parametric models have rarely been applied to the SL domain. Kratimenos et al. (2021) use off-the-shelf methods to estimate a holistic 3D mesh on existing dataset (Theodorakis et al., 2014) but cannot deal with the challenging occlusions and interactions, making them unsuitable for complex, real scenarios. SignBERT+ (Hu et al., 2023) proposed the first self-supervised pre-trainable framework with model-aware hand prior for sign language understanding (SLU). The latest concurrent work (Forte et al., 2023) can reconstruct 3D holistic mesh for SL videos using linguistic priors with group labels obtained from a sign-classifier trained on Corpus-based Dictionary of Polish Sign Language (CDPSL) (Linde-Usiekniewicz et al.), which is annotated with HamNoSys As such, it utilizes an existing sentence segmentation methods (Renz et al., 2021) to generalize to multiple-sign videos. Overall, the literature lacks a robust yet generic method handling **continuous and co-articulated** SL videos with complex hand interactions. **SL datasets**.

While there have been many well-organized continuous 2D SL motion datasets (Duarte et al., 2021; Albanie et al., 2021; 2020; Camgoz et al., 2018; Hanke et al., 2020; Huang et al., 2018), the only existing 3D SL motion dataset with 3D holistic mesh annotation is in (Forte et al., 2023). As mentioned, this rather small dataset only includes a single sign per video only with HamNoSys-prompts. In contrast, SignAvatars provides a **multi-prompt 3D** SL holistic motion dataset with **continuous and co-articulated** signs and fine-grained hand mesh annotations.

**SL applications**. Arkushin et al. (2023) can generate 2D motion sequences from HamNoSys. Saunders et al. (2020b) and Saunders et al. (2021b) are able to generate 3D keypoint sequences relying on glosses. The avatar approaches are often hand-crafted and produce robotic and unnatural movements. Apart from them, there are also early avatar approaches (Ebling & Glauert, 2016; Efthimiou et al., 2010; Bangham et al., 2000; Zwitserlood et al., 2004; Gibet et al., 2016) with a pre-defined protocol and character. To the best of our knowledge, we present the first large-scale 3D holistic SL motion dataset, SignAvatars. Built upon the dataset, we also introduce the novel task and benchmark of 3D sign language production, through different prompts (language, word, HamNoSys).

## 3 SIGNAVATARS DATASET

**Overview**. SignAvatars is a holistic motion dataset composed of $70K$ video clips having $8.34M$ frames in total, containing body, hand and face motions as summarized in Tab. 2. We compile SignAvatars by gathering various data sources from public datasets to online videos and form seven subsets, whose distribution is reported in Fig. 2. Since the individual subsets do not naturally contain expressive 3D whole-body motion labels and 2D keypoints, we introduce a unified automatic annotation framework providing rich 3D holistic parametric SMPL-X annotations along with MANO subsets for hands. Overall, we provide 117 hours of $70K$ video clips with $8.34M$ frames of motion data with accurate expressive holistic 3D mesh as motion annotations.

| Data | Video | Frame | Duration (hours) | Co-articulated | Pose Annotation (to date) | Signer |
|---|---|---|---|---|---|---|
| RWTH-Phoenix-2014T (Camgoz et al., 2018) | 8.25K | 0.94M | 11 | C | - | 9 |
| DGS Corpus (Hanke et al., 2020) | - | - | 50 | C | 2D keypoints | 327 |
| BSL Corpus (Schembri et al., 2013) | - | - | 125 | C | - | 249 |
| MS-ASL (Joze & Koller, 2018) | 25K | - | 25 | I | - | 222 |
| WL-ASL (Li et al., 2020) | 21K | 1.39M | 14 | I | 2D keypoints | 119 |
| How2Sign (Duarte et al., 2021) | 34K | 5.7M | 79 | C | 2D keypoints, depth* | 11 |
| CSL-Daily (Huang et al., 2018) | 21K | - | 23 | C | 2D keypoints, depth | 10 |
| SIGNUM (Von Agris et al., 2008) | 33K | - | 55 | C | - | 25 |
| AUTSL (Sincan & Keles, 2020) | 38K | - | 21 | I | depth | 43 |
| Forte et al. (2023) | 0.05K | 4K | - | I | body mesh vertices | - |
| SignAvatars (Ours) | 70K | 8.34M | 117 | Both | SMPL-X, MANO, 2D&3D keypoints | 153 |

Table 1: Modalities of **publicly available** sign language datasets. C, I represent isolated and co-articulated (continuous) separately. * means the annotation has not been released yet. To the best of our knowledge, our dataset is the first publicly available 3D SL holistic continuous motion dataset with whole-body and hand mesh annotations with the most parallel modalities.

## 3.1 DATASET CHARACTERISTICS

**Expressive motion representation**. To fill in the gaps of previous 2D-only SL data. Our expressive 3D holistic body annotation consists of face, hands, and body, which is achieved by adopting SMPL-X (Pavlakos et al., 2019). It uses standard vertex-based linear blend skinning with learned corrective blend shapes and has N = 10475 vertices and K = 67 joints. For time interval $[1 : t]$, $V_{1:T} = (v_1, ..., v_t)$, $J_{1:T} = (j_1, ..., j_t)$, $\theta_{1:T} = (\theta_1, ..., \theta_t)$, represent mesh vertices, 3d joints, and poses in 6D representation (Zhu et al., 2019). Here the pose $\theta_t$ includes the body pose $\theta_t^b \in R^{23 \times 6}$ with global orientation and the hand pose $\theta_t^h \in R^{30 \times 6}$. Moreover, $\theta_t^f \in R^6$ and $\phi$ represents the yaw pose and facial expressions respectively. For each of the sequences, we use an optimized consistent shape parameter $\tilde{\beta}$ as there is no signer change in each clip. Overall, a motion state $M_t$ is represented as: $M_t = (\theta_t^b, \theta_t^h, \theta_t^f, \phi, \tilde{\beta})$. Moreover, as shown in Tab. 1, our dataset also provides a hand motion subset by replacing the parametric representation from SMPL-X to MANO (Romero et al., 2022): $M_t^h = (\theta_t^h, \tilde{\beta})$, where $h$ is the *handed-ness* and $\tilde{\beta}$ is also an optimized consistent shape parameter.

**Sign language notation**. Similar to spoken languages, sign languages have special structures with a set of linguistic rules (Blaisel, 1997) (*e.g.* grammar, lexicons). Unlike spoken languages, they have no standard written forms. Moreover, there are over 300 different sign languages across the world, with Deaf and hard-of-hearing people who do not know any SL. Hence, having only a single type of annotation is insufficient in practice. To enable more generic applications for different users, we provide more modalities in the SignAvatars dataset. Our SL annotations can be categorized into four common types: HamNoSys, spoken language, word, and gloss, which can be used for a variety of downstream applications such as SLP and SLR.

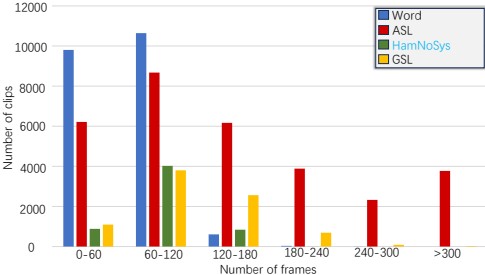

Figure 2: Distribution of subsets. The number of frames for each clip in different subsets. PJM, LSF, DGS and GSL are gathered in one group.

| Data | Video | Frame | Type | Signer |
|---|---|---|---|---|
| Word | 21K | 1.39M | W | 119 |
| PJM | 2.6K | 0.21M | H | 2 |
| DGS | 1.9K | 0.12M | H | 8 |
| GRSL | 0.8K | 0.06M | H | 2 |
| LSF | 0.4K | 0.03M | H | 2 |
| ASL | 34K | 5.7M | S | 11 |
| GSL | 8.3K | 0.83M | S, SG | 9 |
| Ours | 70K | 8.34M | S, H, W, SG | 153 |

Table 2: Statistics of data sources. W, H, S, SG represent **w**ord, **H**amNoSys, sentence-level **s**poken language and sentence-level **g**loss.

**Data sources**. As shown in Tab. 2, SignAvatars leverages our unified automatic annotation framework to collect SL motion sequences in diverse modalities from various different sources. Specifically, for co-articulated SL datasets like How2Sign (Duarte et al., 2021) and How2 (Sanabria et al., 2018) with American Sign Language (ASL) transcriptions, we collect *sentence-level* clips from the *Green Screen studio* subset with multi-view frames, resulting in 34K clips for the **ASL** subset. For **GSL** subset, we mostly gathered data from the publicly available PHOENIX14T dataset (Camgoz et al., 2018) following the official split to have 8.25K video clips. For **HamNoSys** subset, we collect 5.8K isolated-sign SL video clips from Polish SL corpus (Linde-Usiekniewicz et al., 2014)

for PJM, and German Sign Language (DGS), Greek Sign Language (GRSL) and French Sign Language (LSF) from DGS Corpus (Prillwitz et al., 2008) and Dicta-Sign (Matthes et al., 2012). We finally gathered $21K$ clips from word-level sources such as WLASL (Li et al., 2020) to curate the isolated-sign word subset. Overall, we divide our dataset into four subsets: (i)) word, (ii) ASL, (iii) HamNoSys, (iv) GSL based on the prompt categories as shown in Fig. 2.

## 3.2 AUTOMATIC HOLISTIC ANNOTATION

To efficiently auto-label the SL videos with motion data given only RGB online videos, we design an automatic 3D SL annotation pipeline that is not limited to isolated signs. To ensure motion stability and 3D shape accuracy, while maintaining efficiency during holistic 3D mesh recovery from SL videos, we propose an iterative fitting algorithm minimizing an objective heavily regularized both holistically and by *biomechanical hand constraints* (Spurr et al., 2020):

$$E(\theta, \beta, \phi) = \lambda_J L_J + \lambda_\theta L_\theta + \lambda_\alpha L_\alpha + \lambda_\beta L_\beta + \lambda_{\text{smooth}} L_{\text{smooth}} + \lambda_{\text{angle}} L_{\text{angle}} + L_{\text{bio}} \quad (1)$$

where $\theta$ is the full set of optimizable pose parameters, and $\phi$ is the facial expression. $L_J$ represents the joint loss of 2D re-projection, which optimizes the difference between joints extracted from the SMPL-X model, projected into the image, with joints predicted with ViTPose (Xu et al., 2022) and MediaPipe (Kartynnik et al., 2019). The 3D joint can be jointly optimized in $L_J$ when GT is available. $L_\theta$ is the pose prior term following SMPLify-X (Pavlakos et al., 2019). Moreover, $L_\alpha$ is a prior penalizing extreme bending only for elbows and knees and $L_\beta$ is the shape prior term. In addition, $L_{\text{smooth}}$, $L_{\text{angle}}$ and $L_{\text{bio}}$ are the smooth-regularization loss, angle loss and biomechanical constraints, separately. Finally, each $\lambda$ denotes the influence weight of each loss term. Please refer to the appendix for more details. In what follows, we describe in detail our regularizers.

**Holistic regularization.** In terms of reasonable regularization terms, to reduce the jittery results caused by the noisy 2D detected keypoints, we first introduce a smooth term to avoid jitter results for body and hand motion poses, which is defined as:

$$L_{\text{smooth}} = \sum_t (||\hat{\theta}^b_{1:T}||_2 + ||\tilde{\theta}^h_{1:T}||_2 + ||\theta^h_{2:T} - \theta^h_{1:T-1}||_2 + ||\theta^b_{2:T} - \theta^b_{1:T-1}||_2) \quad (2)$$

where $\hat{\theta}^b_{1:T} \in R^{Nxj_b x3}$ is the selected subset of pose parameters from $\theta^b_{1:T} \in R^{NxJx3}$, and N is the frame number of the video. $\tilde{\theta}^h \in R^{Nxj_h}$ is the selected subset of hand parameters from $\theta^b_{1:T} \in R^{NxJ*3}$. The $j_b$ and $j_h$ are the numbers of selected body joint number and hand parameter number, Moreover, this could prevent implausible poses along the bone direction such as twist angle.

After that, we add an angle limit prior term to penalize the hand pose lies outside the plausible range:

$$L_{\text{angle}} = \sum_t (\mathcal{I}(||\theta^h_{1:T}||_2; \theta^h_{\min}, \theta^h_{\max}) + \mathcal{I}(||\theta^b_{1:T}||_2; \theta^b_{\min}, \theta^b_{\max})) \quad (3)$$

where $\mathcal{I}$ is the interval loss penalizing the outliers, $\theta^{h,b}_{\min}, \theta^{h,b}_{\max}$ is the pre-defined interval, $\theta^h, \theta^b$ is the selected subset of holistic poses. Finally, the signer in each video clip will not change, so we can use an **optimized consistent shape parameters** $\beta$ to represent the holistic body shape. Specifically, our fitting procedure is split into **five** stages, where we will optimize the shape for the first **three** stages of optimization to derive the mean shape and freeze the shape in the following stages.

**Biomechanical hand constraints**. Hand pose estimation from monocular RGB images is challenging due to fast movements, interaction, frequent occlusion and confusion. To further improve the hand motion quality and eliminate implausible hand pose. We apply biomechanical constraints to our hands, which contain three terms: (i) $L_{\text{bl}}$ for bone length, (ii) $L_{\text{palm}}$ for palmar region optimization, and (iii) $L_{\text{ja}}$ for joint angle priors. Specifically, $L_{\text{bio}}$ is defined as the weighted sum of:

$$L_{\text{bl}} = \sum_i \mathcal{I}(||b^i_{1:T}||_2; b^i_{min}, b^i_{max}), \quad L_{\text{ja}} = \sum_i D(\alpha^i_{1:T}, H^i)$$

$$L_{\text{palm}} = \sum_i (\mathcal{I}(||c^i_{1:T}||_2; c^i_{min}, c^i_{max}) + \mathcal{I}(||d^i_{1:T}||_2; d^i_{\min}, d^i_{\max})), \quad (4)$$

where $\mathcal{I}$ is the interval loss penalizing the outliers, $b_i$ is the bone length of $i$-th finger bone and the optimization constraints the whole sequence $[1 : T]$. After that, we further constrain the curvature

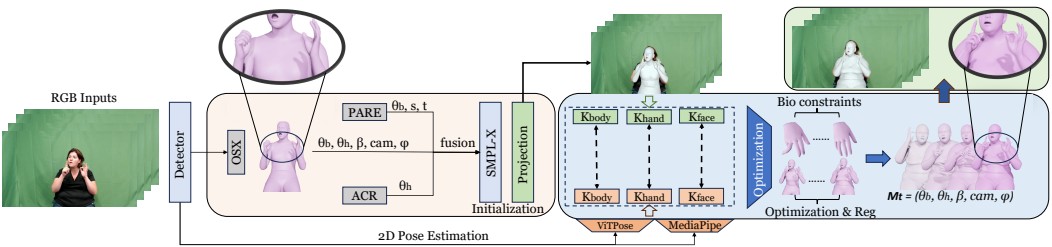

Figure 3: Overview of our automatic annotation pipeline. Given an RGB image sequence as input for hierarchical initialization, it is followed by optimization with temporal smoothing and biomechanical constraints. Finally, it outputs the final results in a motion sequence of SMPL-X parameters.

and angular distance for the 4 root bones of palmar structures by penalizing the outliers of curvature range $c_{max}^i, c_{min}^i$ and angular distance range $d_{\max}^i, d_{\min}^i$. Inspired by Spurr et al. (2020), we also apply constraints to the sequence of joint angles $\alpha_{1:T}^i = (\alpha_{1:T}^f, \alpha_{1:T}^a)$ by approximating the convex hull on $(\alpha^f, \alpha^a)$ plane with point set $H^i$ and minimizing their distance $D$, where $(\alpha^f, \alpha^a)$ is the flexion and abduction angles. The biomechanical loss is then computed as the weighted sum of them: $L_{bio} = \lambda_{bl}L_{bl} + \lambda_{palm}L_{palm} + \lambda_{ja}L_{ja}$. We refer the reader to our appendix for more details.

**Hierarchical initialization**. Given an RGB image sequence, we initialize the holistic SMPL-X parameters from OSX (Lin et al., 2023b). Though, due to the frequent occlusion and hand interactions, OSX is not always sufficient for a good initialization. Therefore, we further fuse OSX with ACR (Yu et al., 2023), PARE (Kocabas et al., 2021) to improve stability under occlusion and truncation. For 2D holistic keypoints initialization, we first train a whole-body 2D pose estimation model on COCO-wholeBody (Jin et al., 2020) based on ViTPose (Xu et al., 2022) and subsequently incorporated with MediaPipe (Kartynnik et al., 2019) by fusing and feeding through a confidence-guided filter.

## 4 APPLICATIONS: SIGNVAE

Our SignAvatars dataset enables the first applications to generate high-quality and natural 3D sign language holistic motion along with 3D meshes from both isolated and continuous SL prompts. To this end, motivated by the fact that the text prompts are highly correlated and aligned with the motion sequence, our method consists of a two-stage process designed to enhance the understanding of varied inputs by focusing on both semantic and motion aspects. In the first stage, we develop two codebooks - a shared semantic codebook and a motion codebook - by employing two Vector Quantized Variational Auto-Encoders (VQ-VAE). This allows us to map diverse inputs to their corresponding semantic code indices and link motion elements to motion code indices. In the second stage, we utilize an auto-regressive model to generate motion code indices based on the previously determined semantic code indices. This integrated approach ensures a coherent and logical understanding of the input data, effectively capturing both the semantic and motion-related information.

**SL motion generation**. To produce stable and natural holistic poses, instead of directly mapping prompts to motion, we leverage the generative model VQ-VAE as our SL motion generator. As illustrated in Fig. 4, our SL motion VQVAE consists of an autoencoder structure and a learnable codebook $Z_m$, which contains $I$ codes $Z_m = \{z_i\}_{i=1}^I$ with $z_i \in R^{d_z}$. We first encode the given 3D SL motion sequence $M_{1:T} = (\theta_T^b, \theta_T^h, \theta_T^f, \phi)$, where $T$ is the motion length, into a latent feature $F_{1:(T/w)}^m = (f_1^m, ..., f_{1:(T/w)}^m) \in R^{d_z}$, where $w = 4$ is used as the downsampling rate for the window size. Subsequently, we quantize the latent feature embedding by searching for the nearest neighbour code in the codebook $Z_m$. For the $j-th$ feature, the quantization code is found by: $\hat{f}_j^m = \arg\min_{z_i \in Z} ||f_j^m - z_i||_2$. Finally, the quantized latent features are fed into decoders for reconstruction.

In terms of the training of the SL motion generator, we apply the standard optimization scheme with $L_{motion_vq}$:

$$L_{m-vq} = L_{recon}(M_{1:T}, \hat{M}_{1:T}) + ||sg[F_{1:T}^m] - F_{1:T}^{\hat{m}}||_2 + \beta||F_{1:T}^m - sg[F_{1:T}^{\hat{m}}]||_2 \qquad (5)$$

where $L_{recon}$ is the MSE loss and $\beta$ is a hyper-parameter. sg is the detach operation to stop the gradient. We provide more details regarding the network architecture and training in our appendix.

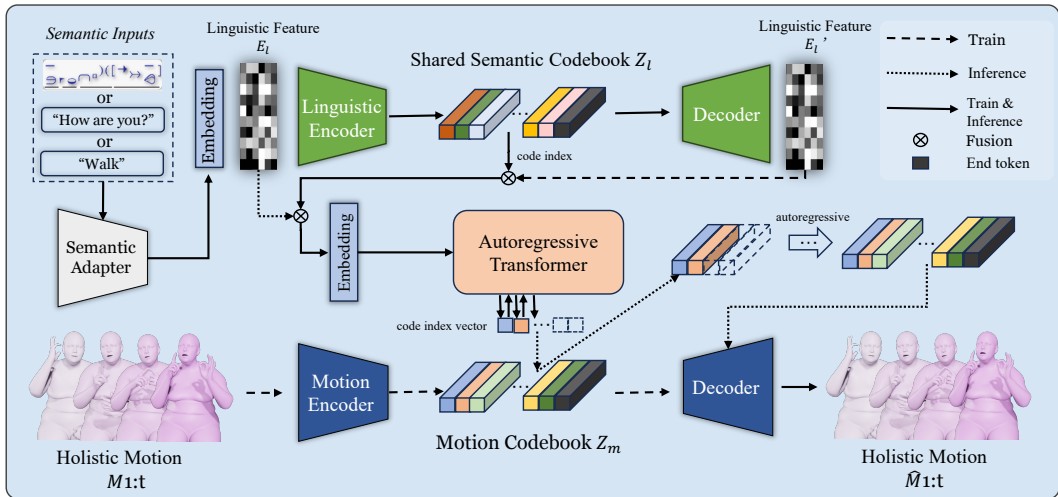

Figure 4: Overview of our 3D SLP network. Our method consists of a two-stage process. We first create semantic and motion codebooks using two VQ-VAEs, mapping inputs to their respective code indices. Then, we employ an auto-regressive model to generate motion code indices based on semantic code indices, ensuring a coherent understanding of the data.

**Prompt feature extraction for parallel linguistic feature generation..** In terms of the linguistic condition $c$ from input prompt, typical motion generation tasks usually leverage LLM to produce linguistic prior for efficient learning. In our task of spoken language and word-level annotation, we leverage CLIP (Radford et al., 2021) as our prompt encoder to obtain the text embedding $E^l$. However, this does not extend to all the other SL annotations we desire. As a remedy, to enable applications with different prompts such as HamNoSys, instead of relying on the existing pre-trained CLIP, we define a new prompt encoder for embedding. After quantizing the prompt (*e.g.* HamNoSys glyph) into tokens with length $s$, we use an embedding layer to produce the linguistic feature $\hat{E}^l_{1:s} = (\hat{e}^l_1, ..., \hat{e}^l_s)$ with same dimension $d_l$ as the text embeddings of CLIP (Radford et al., 2021). For simplicity, we use "text" to represent all different input prompts. Subsequently, motivated by the fact that the text prompts are highly correlated and aligned with the motion sequence, we propose a linguistic VQVAE as our *parallel linguistic feature generator* (PLFG) module coupled with the SL motion generator. In particular, we leverage a similar quantization process using the codebook $Z_l$ and training scheme as in the SL motion generator to yield linguistic features:

$$L_{l-vq} = L_{\text{recon}}(E^l_{1:s}, \hat{E}^l_{1:s}) + ||sg[F^l_{1:s} - \hat{F^l}_{1:s}||_2 + \beta||F^l_{1:s} - sg[\hat{F^l}_{1:s}]||_2 \qquad (6)$$

where $F^l_{1:s}$ is the latent feature after encoding the initial linguistic feature. $\hat{F^l}_{1:s}$ is the quantized linguistic feature after applying $\hat{f}^l_j = \arg\max_{z_i \in Z_l} ||f^l_j - z_i||_2$ to $F^l_{1:s}$.

**Sign-motion cross modelling and production**. After training the VQVAE-based SL motion generator, we can map any motion sequence $M_{1:T}$ to a sequence of indices $X = [x_1, ..., x_{T/w}, x_{\text{EOS}}]$ through the motion encoder and quantization, where $end$ is a learnable end token representing the *stop* signal. After training both the SL motion generator and the linguistic feature generator, our network will be jointly optimized in a parallel manner. Specifically, we fuse the linguistic feature embedding $E_l$ and the codebook index vectors of $Z_l$ to formulate the final condition for our autoregressive code index generator. The objective for training the code index generator can be seen as an autoregressive next-index prediction task, represented with a $\text{cross} - \text{entropy}$ loss between the likelihood of the full predicted code index sequence and the real ones as $L_{\text{SLP}} = \mathbb{E}_{X \sim p(X)}[-\log p(X|c)]$.

Lastly, with the quantized motion representation, we generate the codebook vectors in a temporal autoregressive manner and predict the distribution of the next codebook indices given an input linguistic prompt as linguistic condition $c$. After mapping the codebook indices $X$ to the quantized motion representation $\hat{F}^m_{1:(T/w)}$, we are able to decode and produce the final 3D holistic motion with mesh representations $M_{1:T}$.

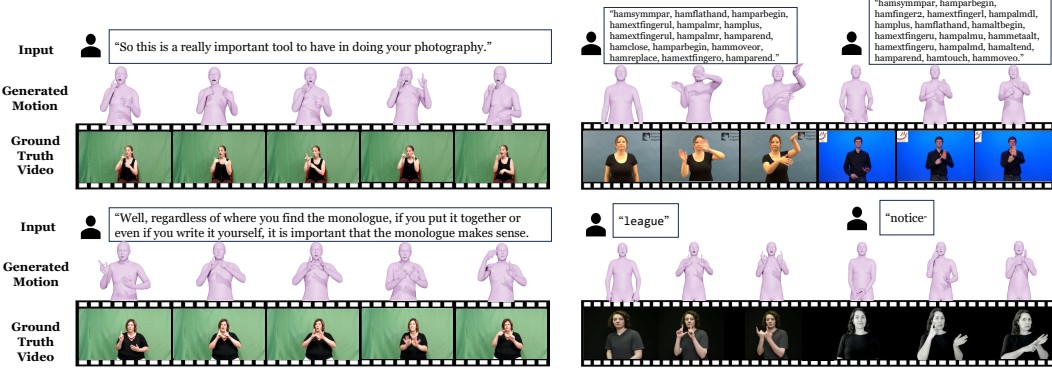

Figure 5: Qualitative results of 3D holistic SLP from different prompts (left row: spoken language, top right: HamNoSys, bottom right: word). Within each sample, the first two rows are the input prompts and the generated results. The last row is the corresponding video clip from our dataset.

## 5 EXPERIMENTAL EVALUATION

With SignAvatars dataset, we have enabled more 3D applications for sign language communities, especially 3D SL motion production. We now showcase the effectiveness and contribution of SignAvatars on the benchmark and application introduced in Sec 4. Note that, these are also thee first benchmark results for 3D holistic SL motion production yielding mesh representations.

**Evaluation metrics**. To fully assess the quality of our motion generation, we evaluate the holistic motion as well as the arm motion[1]. Based on an evaluation model trained following prior arts in motion generation (Tevet et al., 2022; Zhang et al., 2023b), we use the scores and metrics of FID, Diversity, Multimodality (MM), MM-Dist, MR-precision, whose details are provided in App. C.1. Unfortunately, there is no de-facto standard for evaluating 3D SLP in the literature. While Lee et al. (2023) is capable of back-translating 3D SL motion by treating it as a classification, it is tailored only for word-level back-translation. While BLEU and ROUGE are commonly used in the back-translation evaluation (Saunders et al., 2020b; 2021b), they are not generic for other types of annotations such as HamNoSys or glosses. Since the generated motion might differ in length from the real motion, absolute metrics like MPJPE would also be unsuited. Inspired by (Arkushin et al., 2023; Huang et al., 2021), we propose a new **MR-Precision** for motion retrieval as well as DTW-MJE (Dynamic Time Warping - Mean Joint Error) (Kruskal, 1983) with standard SMPL-X keypoint set without lower body, for evaluating the performance of our method as well as the baselines.

**Subsets & training settings.**. Specifically, we report results on three representative subsets: (i) the complete set of $ASL$ for spoken language (corresponding to $language$ in Table 3), (ii) the $word$ subset with 300 vocabularies, (iii) combined subset of DGS, LSF, PJM, and GRSL for HamNoSys. For training, we follow the official splits for (i) and (ii). For (iii), we leverage a four-fold strategy where we train on three of them and test on the other, repeated four times to have the final results.

**Qualitative analysis**. Fig. 5 shows examples of our 3D holistic body motion generation results. As observed, our method can generate plausible and accurate holistic 3D motion from different prompts while containing some diversity enriching the production results. We show further comparisons against the state-of-the-arts in Appendix.

**Benchmarking & quantitative analysis**. To the best of our knowledge, there is no publicly available benchmark for 3D mesh & motion-based SLP[2]. To evaluate SignAvatars as the first 3D motion-based SLP benchmark, we present detailed quantitative results in Tab. 3. It can be seen that the 3D SLP with word-level prompts can achieve the best performance reaching the quality of real motions. Learning from spoken languages is a naturally harder task and we invite the community to develop stronger methods to produce 3D SLP from spoken languages. To further evaluate the sign accuracy and effect of body movement, we report separate results for individual arms (*e.g.* "$Gesture$"), with slight improvements in FID and MR-Precision. However, it will also degenerate the text-motion consistency (R-Precision and MM-dist) due to the absence of body-relative hand position.

---

[1]The lower body is not evaluated in our experiments as it is not related to the SL motion.

[2]Saunders et al. (2020a) does not provide a public evaluation model as discussed in our Appendix.

| Data Type | | R-Precision (↑) | | | FID (↓) | Diversity (→) | MM (→) | MM-dist (↓) | MR-Precision (↑) | | |
|---|---|---|---|---|---|---|---|---|---|---|---|
| | | top 1 | top 3 | top 5 | | | | | top 1 | top 3 | top 5 |
| **Real motion** | Language | $0.375^{\pm.005}$ | $0.545^{\pm.007}$ | $0.679^{\pm.008}$ | $0.061^{\pm.153}$ | $12.11^{\pm.075}$ | - | $3.786^{\pm.057}$ | - | - | - |
| | HamNoSys | $0.455^{\pm.002}$ | $0.689^{\pm.006}$ | $0.795^{\pm.004}$ | $0.007^{\pm.062}$ | $8.754^{\pm.028}$ | - | $2.113^{\pm.023}$ | - | - | - |
| | Word-300 | $0.499^{\pm.003}$ | $0.811^{\pm.002}$ | $0.865^{\pm.003}$ | $0.006^{\pm.054}$ | $8.656^{\pm.035}$ | - | $1.855^{\pm.019}$ | - | - | - |
| Holistic | Language | $0.265^{\pm.007}$ | $0.413^{\pm.008}$ | $0.531^{\pm.0059}$ | $4.359^{\pm.389}$ | $12.35^{\pm.101}$ | $3.451^{\pm.107}$ | $4.851^{\pm.067}$ | $0.356^{\pm.007}$ | $0.525^{\pm.007}$ | $0.645^{\pm.009}$ |
| | HamNoSys | $0.429^{\pm.004}$ | $0.657^{\pm.005}$ | $0.756^{\pm.002}$ | $0.884^{\pm.035}$ | $9.451^{\pm.087}$ | $0.941^{\pm.056}$ | $2.651^{\pm.027}$ | $0.552^{\pm.002}$ | $0.745^{\pm.010}$ | $0.813^{\pm.034}$ |
| | Word-300 | $0.475^{\pm.002}$ | $0.731^{\pm.003}$ | $0.815^{\pm.005}$ | $0.756^{\pm.021}$ | $8.956^{\pm.091}$ | $0.815^{\pm.059}$ | $2.101^{\pm.024}$ | $0.615^{\pm.005}$ | $0.797^{\pm.006}$ | $0.875^{\pm.002}$ |
| Gesture | Language | $0.245^{\pm.008}$ | $0.405^{\pm.009}$ | $0.519^{\pm.010}$ | $3.951^{\pm.315}$ | $10.12^{\pm.121}$ | $3.112^{\pm.135}$ | $5.015^{\pm.089}$ | $0.375^{\pm.011}$ | $0.535^{\pm003}$ | $0.668^{\pm.004}$ |
| | HamNoSys | $0.435^{\pm.005}$ | $0.649^{\pm.004}$ | $0.745^{\pm.006}$ | $0.851^{\pm.033}$ | $8.944^{\pm.097}$ | $0.913^{\pm.036}$ | $2.876^{\pm.015}$ | $0.581^{\pm.004}$ | $0.736^{\pm.006}$ | $0.825^{\pm.008}$ |
| | Word-300 | $0.465^{\pm.001}$ | $0.711^{\pm.003}$ | $0.818^{\pm.003}$ | $0.715^{\pm.016}$ | $8.235^{\pm.055}$ | $0.801^{\pm.021}$ | $2.339^{\pm.027}$ | $0.593^{\pm.006}$ | $0.814^{\pm.005}$ | $0.901^{\pm.006}$ |

Table 3: Quantitative evaluation results for the 3D holistic SL motion generation. *Real motion* is the sampled motions from the original holistic motion annotation in the dataset. *Holistic* represents the results for generated holistic motion. *Gesture* stands for the evaluation conducted on two arms.

Due to the lack of works that can generate 3D holistic SL motion with mesh representation from any of the linguistic sources (*e.g.* spoken language, HamNoSys, gloss, ...), we modify the latest HamNoSys-based SLP work, Ham2Pose (Arkushin et al., 2023) (*Ham2Pose-3d* in Tab. 4), as well as MDM (Tevet et al., 2022) (corresponding to *SignDiffuse* in Tab. 5), to take our linguistic feature as input and to output SMPL-X representations and evaluate on our dataset. We theen train our SignVAE and *Ham2Pose-3d* along with the original *Ham2Pose* on their official split and use DTW-MJE for evaluation. Specifically, we also regress the keypoints from our holistic representation $M_{1:T}$ to align with the Ham2Pose 2D skeleton. As discovered in this benchmark, leveraging our SignAvatars dataset can easily enable more 3D approaches and significantly improve the existing SLP applications by simple adaptation compared to the original Ham2Pose. The results in Tab. 4 are reported on the HamNoSys *holistic* set for comparison. While our method drastically improves over the baseline, the result is far from ideal, motivating the need for better models for this new task.

| Method | DTW-MJE Rank (↑) | | |
|---|---|---|---|
| | top 1 | top 3 | top 5 |
| Ham2Pose* | $0.092^{\pm.031}$ | $0.197^{\pm.029}$ | $0.354^{\pm.032}$ |
| Ham2Pose-3d | $0.253^{\pm.036}$ | $0.369^{\pm.039}$ | $0.511^{\pm.035}$ |
| SignVAE (Ours) | $\mathbf{0.516}^{\pm.039}$ | $\mathbf{0.694}^{\pm.041}$ | $\mathbf{0.786}^{\pm.035}$ |

Table 4: Comparison with state-of-the-art SLP method from HamNoSys holistic subset. * represents using only 2D information.

| Method | R-Precision (↑) | | | MM-dist (↓) |
|---|---|---|---|---|
| | top 1 | top 3 | top 5 | |
| Ham2Pose-3d | $0.291^{\pm.003}$ | $0.386^{\pm.005}$ | $0.535^{\pm.005}$ | $3.875^{\pm.086}$ |
| SignDiffuse | $0.285^{\pm.003}$ | $0.415^{\pm.005}$ | $0.654^{\pm.003}$ | $3.866^{\pm.054}$ |
| SignVAE (Base) | $0.385^{\pm.008}$ | $0.613^{\pm.009}$ | $0.745^{\pm.007}$ | $3.056^{\pm.108}$ |
| SignVAE (Ours) | $\mathbf{0.429}^{\pm.009}$ | $\mathbf{0.657}^{\pm.008}$ | $\mathbf{0.756}^{\pm.008}$ | $\mathbf{2.651}^{\pm.119}$ |

Table 5: Quantitative ablation study of Sign-VAE on HamNoSys *holistic* subset for comparison with prior arts.

**Ablation study**. For the study of our unique text-sign cross-modeling module, our baseline, *SignVAE (Base)*, replaces the PLFG with a canonical pre-trained CLIP feature as input to the encoder. As shown in Tab. 5, our joint scheme utilizing the PLFG can significantly improve the prompt-motion consistency, resulting in an increase in **R-precision** and **MM-dist**. Moreover, our VQVAE backbone quantizing the motion representation into a motion codebook, enables interaction with the linguistic feature codebook, leading to significant improvements in prompt-motion correspondences and outperforms other baselines built with our linguistic feature generator (SignDiffuse, Ham2Pose-3d) and generates more text-motion consistent results.

# 6 CONCLUSION

We introduced **SignAvatars**, the first large-scale 3D holistic SL motion dataset with expressive 3D human and hand mesh annotations, provided by our automatic annotation pipeline. SignAvatars enables a variety of application potentials for Deaf and hard-of-hearing communities. Built upon our dataset, we propose the first 3D sign language production approach to generate natural holistic mesh motion sequences from SL prompts. We also introduce the first benchmark results for this new task, 3D holistic SL motion production from diverse SL prompts. Our evaluations on this benchmark clearly show the advantage of our new VQVAE-based model over the baselines, we develop.

**Limitations and Future Work:** Having the first benchmark we proposed, there is still a lack of in-depth investigation of other 3D techniques for 3D SL motion generation. Especially, due to the lack of sophisticated generic 3D back-translation methods, the evaluation may not fully showcase the superiority of our dataset and the proposed method. We leave this for a future study. Moreover, combining 3D SLT and SLP to formulate a multi-modal generic SL framework will be one of the future works. Developing a large sign language model with more properties and applications in AR/VR will significantly benefit the Deaf and hard-of-hearing people around the world.

ETHICS STATEMENT

Our work is driven by a dedication to the advancement of knowledge and the betterment of society. As an important step towards bringing our 3D digital world to the Deaf community, most importantly, our first publicly available large-scale 3D holistic SL motion dataset will significantly boost the development and novel research ideas of 3D SL applications especially 3D SLP and 3D SLT, which can be used for SL understanding and simultaneous SL virtual signer. Furthermore, our dataset is not only capable of 3D SL research, but it also possesses a good potential for motion capture areas to gain more accurate hand pose, as well as many 3D applications in AR/VR, animation, and human-computer interaction (HCI). Our dataset will follow the license of the collection source and will not present any negative social impact.

All of our experiments were either run on publicly available datasets or on data that is synthetically generated. While these datasets contained human annotations, they have been licensed appropriately. We included these licenses in our appendix. No human or animal subjects have been involved at any stage of the work we were involved. Trained on this data, our models are designed to enhance the understanding and production of sign languages, without causing harm or perpetuating unjust biases, unless provided in the datasets. While we do not foresee any issue with methodological bias, we have not analyzed the inherent biases of our algorithm and there might be implications in applications demanding utmost fairness.

We aptly acknowledge the contributions of researchers whose work laid the foundation for our own. Proper citations and credit are given to previous studies and authors. All authors declare that there are no conflicts of interest that could compromise the impartiality and objectivity of this research. All authors have reviewed and approved the final manuscript before submission.

REPRODUCIBILITY STATEMENT

We are committed to transparency in research and for this reason will make our implementation publicly available upon publication. To demonstrate our dedication, we have submitted all source code as part of the appendix.

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

APPENDIX

## A    ADDITIONAL VISUALIZATIONS OF SIGNAVATARS DATASET

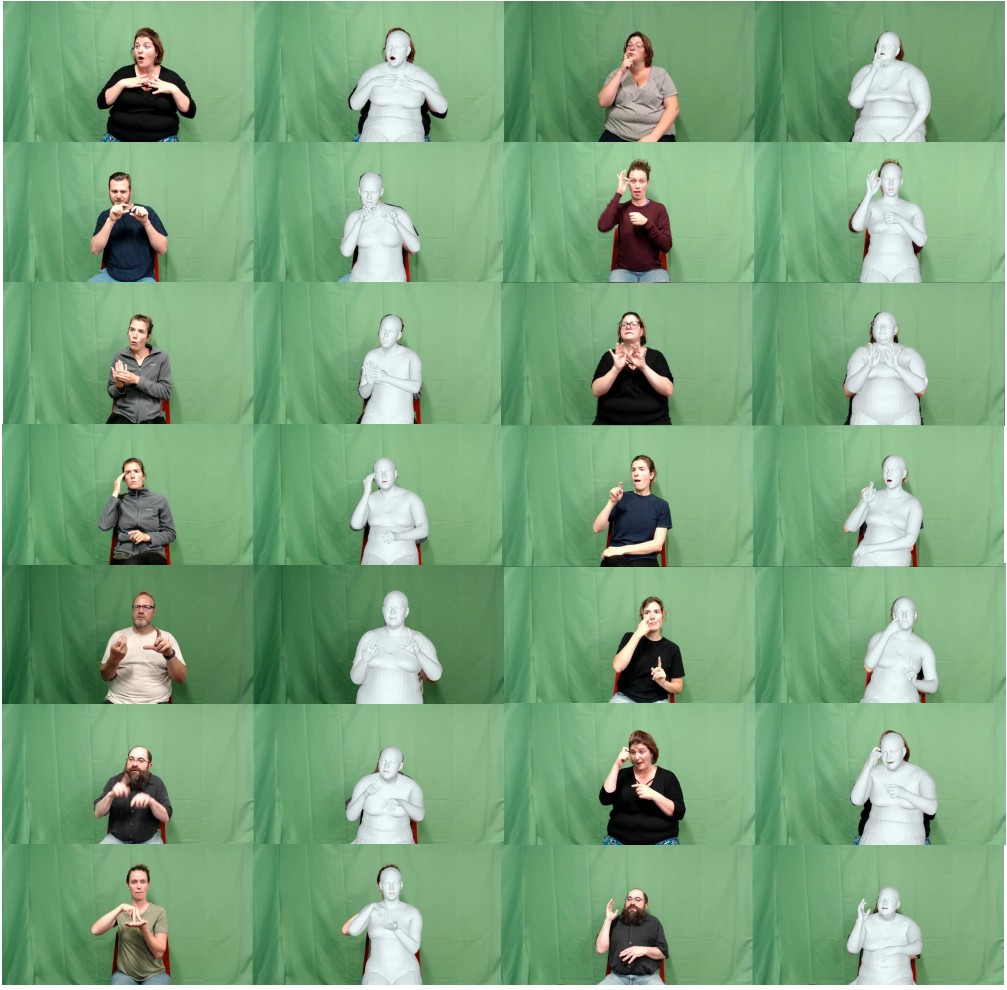

Figure 6: More sentence-level spoken language examples of SignAvatars, $ASL$ subset. We have different shapes of annotations presenting the accurate body and hand estimation.

In this section, we present more samples and visualizations of our SignAvatars dataset for each of the subsets categorized by the annotation type: spoken language (sentence-level), HamNoSys, and word-level prompt annotation.

### A.1    QUALITATIVE ANALYSIS OF SIGNAVATARS DATASET

We provide further details of our SignAvatars dataset and present more visualization of our data in Fig. 6, Fig. 7 and 8. Being the first large-scale multi-prompt 3D sign language (SL) motion dataset with accurate holistic mesh representations, our dataset enables various tasks such as 3D sign language recognition (SLR) and the novel 3D SL production (SLP) from diverse inputs like text scripts, individual words, and HamNoSys notation. We also provide a demo video in the supplementary materials and our anonymous project page: https://anonymoususer4ai.github.io/.

### A.2    MORE GENERATION SAMPLES FROM SIGNVAE

We now share snapshot examples produced from our SignVAE, demonstrating the application potential for 3D sign language production in our demo video on project page.

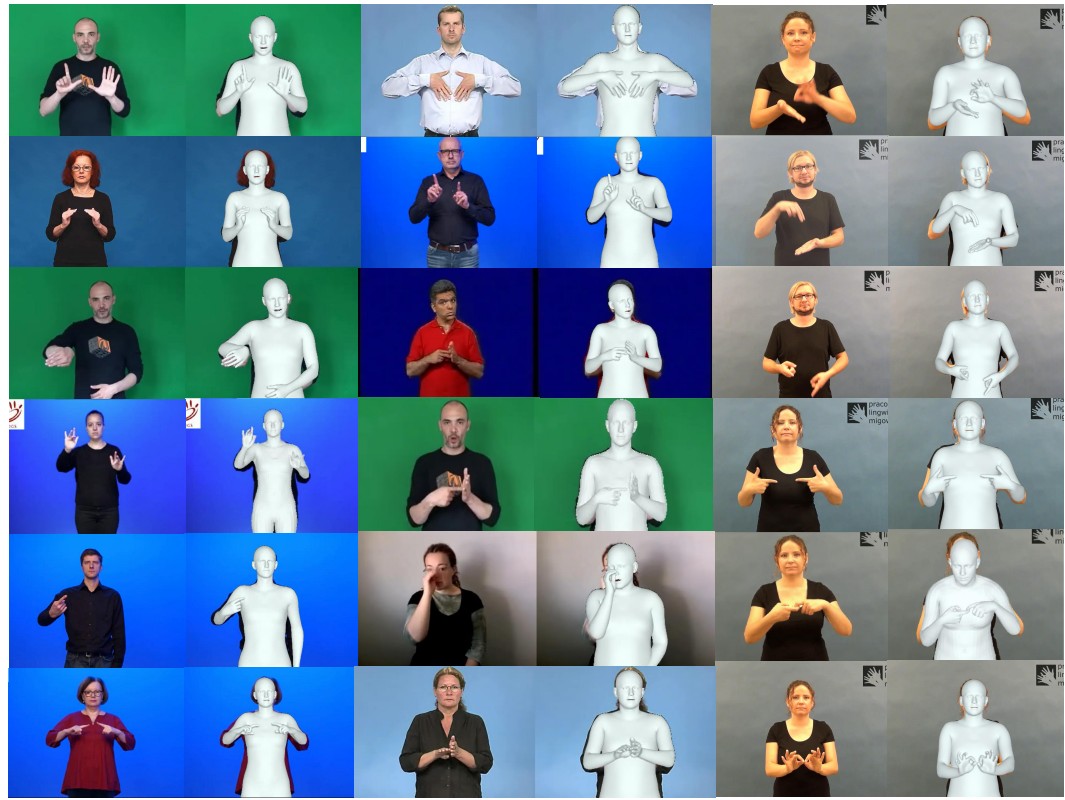

Figure 7: More HamNoSys-level examples of SignAvatars, $HamNoSys$ subset. We have different shapes of annotations presenting the accurate body and hand estimation.

# B  MORE EVALUATION AND EXPERIMENTAL RESULTS

## B.1  EVALUATION OF ANNOTATION PIPELINE

**Comparison with state-of-the-art methods on our dataset**. To showcase the effectiveness of our annotation pipeline and its capability of recovering accurate hand & holistic mesh annotation. We have conducted experiments against previous state-of-the-art monocular methods including SMPLify-x (Pavlakos et al., 2019), OSX (Lin et al., 2023b), PyMAF-X (Zhang et al., 2023a), PIXIE (Feng et al., 2021). As shown in Fig. 9, our method generates significantly more natural body movement, as well as accurate hand pose and better pixel-mesh aligned body shape ($\beta$).

**Comparison with state-of-the-art methods on other datasets**. We evaluate our optimization-based automatic annotation pipeline on the popular EHF dataset (Pavlakos et al., 2019). The qualitative comparison with the state-of-the-art human mesh recovery methods can be found at Fig. 10. For quantitative evaluation, we follow the prior works to apply per-vertex error (MP-VPE), mean per-vertex error after Procrusters alignment (PA-MPVPE), and mean per-joint error after Procrusters alignment (PA-MPJPE). It can be seen from Tab. 6 that our method significantly surpasses the existing popular monocular holistic reconstruction methods by a large margin. Overall, our PA-MPJPE shows an 40% improvement than SOTA Lin et al. (2023a). Specifically, our hand reconstruction error can achieve 4.7 for PA-MPVPE when the biomechanical constraints are employed.

**Effectiveness of regularization and in-the-wild application**. Our method is not limited to SL video. We provide more in-the-wild examples with our annotation methods in Fig. 11. As we can see from the comparison with state-of-the-art methods, our method provides significantly better quality regarding **pixel alignment**, especially with more natural and plausible hand poses. Subsequently, the biomechanical constraints can serve as a prior for eliminating the implausible poses, which happens frequently in complex interacting-hands scenarios for other monocular capture methods, as shown in Fig. 12.

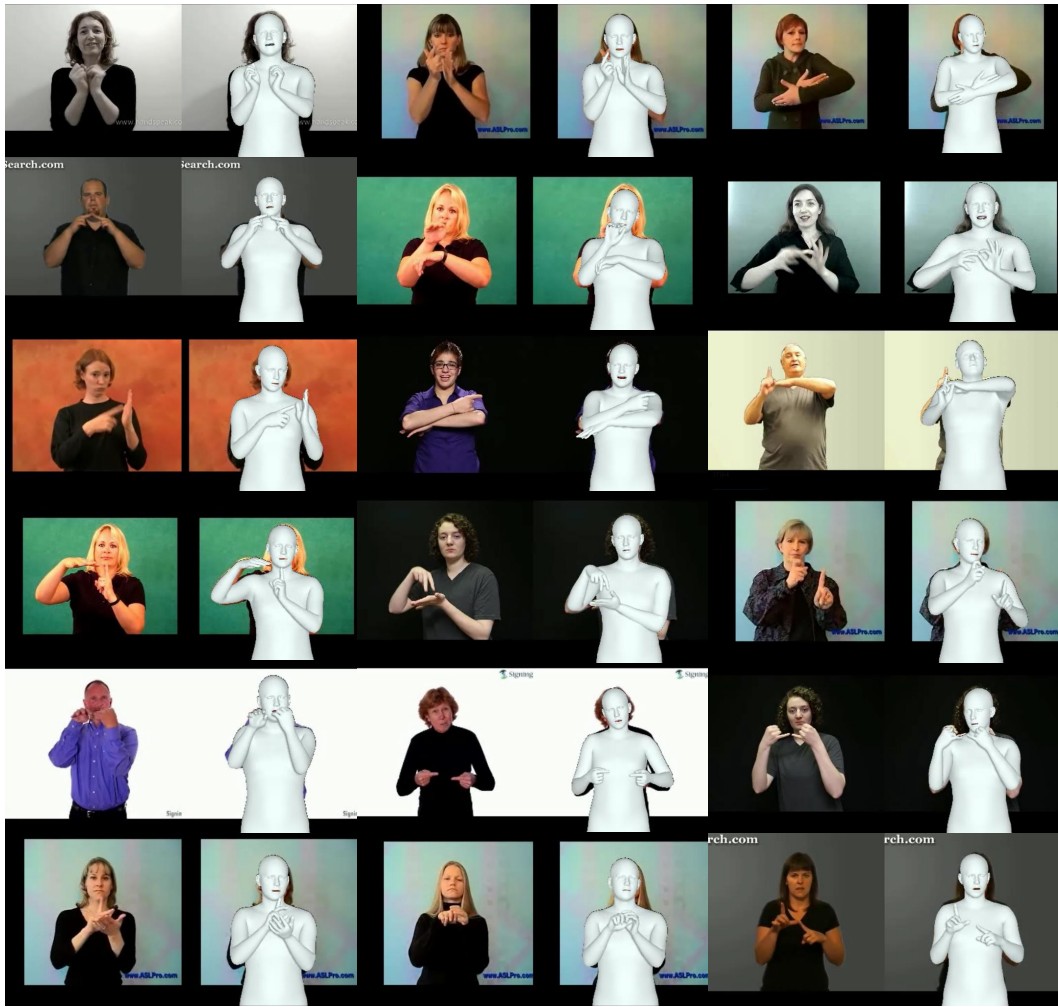

Figure 8: More word-level examples of SignAvatars, *word* subset. We have different shapes of annotations presenting the accurate body and hand estimation.

| Method | MPVPE | | | PA-MPVPE | | | PA-MPJPE | |
|---|---|---|---|---|---|---|---|---|
| | Holistic | Hands | Face | Holistic | Hands | Face | Body | Hands |
| SMPLify-X (Pavlakos et al., 2019)* | - | - | - | 65.3 | 75.4 | 12.3 | 62.6 | 12.9 |
| FrankMocap (Rong et al., 2021)† | 107.6 | 42.8 | - | 57.5 | 12.6 | - | 62.3 | 12.9 |
| PIXIE (Feng et al., 2021)† | 89.2 | 42.8 | 32.7 | 55.0 | 11.1 | 4.6 | 61.5 | 11.6 |
| Hand4Whole (Moon et al., 2022)† | 76.8 | 39.8 | 26.1 | 50.3 | 10.8 | 5.8 | 60.4 | 10.8 |
| PyMAF-X (Feng et al., 2021)† | 64.9 | 29.7 | 19.7 | 50.2 | 10.2 | 5.5 | 52.8 | 10.3 |
| OSX (Lin et al., 2023b)† | 70.8 | - | - | 48.7 | - | - | 55.6 | - |
| Motion-X (Lin et al., 2023a)‡ | 44.7 | - | - | 31.8 | - | - | 33.5 | - |
| Motion-X w/GT 3Dkpt (Lin et al., 2023a)‡ | 30.7 | - | - | 19.7 | - | - | 23.9 | - |
| Ours (w/o bio)* | 21.6 | 12.5 | 7.8 | 14.2 | 5.4 | 4.3 | 16.5 | 6.2 |
| Ours* | **20.1** | **9.7** | **7.8** | **12.9** | **4.7** | **4.3** | **15.6** | **5.8** |

Table 6: Quantitative comparison on EHF dataset, where * denotes the optimization-based approaches, † denotes the regression-based method, and ‡ represents the hybrid method.

## B.2    EVALUATION OF SIGNVAE GENERATION ON OTHER BENCHMARKS

In this section, we aim to conduct further experiments with our SignVAE on 3D SLP from spoken language on other benchmarks to further showcase its ability. To the best of our knowledge, no publicly available benchmark for **3D mesh & motion-based** SLP exists. Progres-

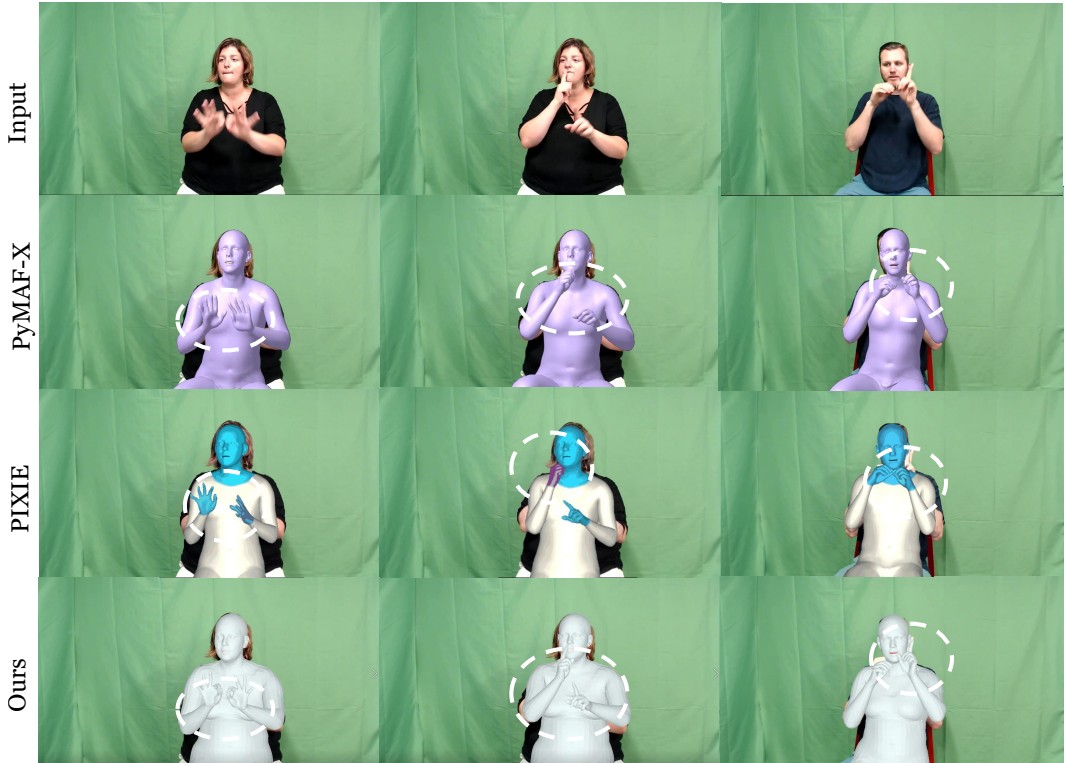

Figure 9: **Comparison of 3D holistic body reconstruction.** The results from PIXIE (Feng et al., 2021), PyMAF-X (Zhang et al., 2023a), and ours.

sive Transformer (Saunders et al., 2020b) and its continuation series (Saunders et al., 2021b;a; 2020a) on RWTH-PHOENIX-Weather 2014 T dataset (Camgoz et al., 2018) provides a **keypoint-based** 3D Text2Pose (Language2Motion) benchmark. Unfortunately, since, at the time of submission, this benchmark was not publicly available. Note that, conducting back-translation evaluations as in (Saunders et al., 2020b) must strictly follow the rule to use the same back-translation model checkpoint for a fair comparison. This is also the same for the human motion generation area, where all the evaluations should be conducted with the same evaluation checkpoints such as the popular HumanML3D benchmark does Guo et al. (2022). Unfortunately, the pre-trained evaluation model checkpoint or its reproductions are available neither on the project website https://github.com/BenSaunders27/ProgressiveTransformersSLP (with an open issue) or on other sites, We have not managed to get in touch with the corresponding authors. For this reason, **we have re-evaluated the benchmark method in (Saunders et al., 2020b)** as follows:

**Experimental Details**. To conduct evaluations on Phoenix-2014T using the Progressive Transformer (PT) (Saunders et al., 2020b), we trained our network as well as PT on this dataset and recorded new results under our metrics. We conduct the re-evaluation experiments by:

- First, we generate mesh annotations for the Phoenix-2014T dataset and add them as our subsets GSL. We follow the original data distribution and official split to train our network.
- Second, because in addition to the absence of the evaluation model, the generation model checkpoints are also lacking, we re-train PT using the official implementation on both 3D-lifted OpenPose keypoints $J_{PT}$ and the 3D keypoints $J_{ours}$ regressed from our mesh representation, corresponding to PT ($J_{PT}$) and PT ($J_{ours}$).
- Third, we train two 3D keypoints-based SL motion evaluation models on this subset with $J_{PT}$ and $J_{ours}$, resulting in two model checkpoints $C_{PT}$ and $C_{ours}$.

**Comparisons**. We conduct both quantitative and qualitative comparisons between the PT and our method, following the official split with both $C_{PT}$ and $C_{ours}$ in Tab. 7 under our evaluation metrics introduced in Sec 5 and App. C.1. As shown in Tab. 7, our method significantly outperforms

PT, especially regarding the R-precision and MR-precision, which indicates better prompt-motion consistency. Moreover, we can discover from the evaluation of Real Motion that the evaluation model $C_{ours}$ utilizing the 3D keypoints $J_{ours}$ regressed from our mesh representation can provide essentially better matching accuracy with less noise (MM-dist) than the noisy canonical 3d-lifted $OpenPose$ keypoints $J_{PT}$, yielding better performance than using $C_{PT}$. A carefully designed evaluation model with proper training data will significantly improve the ability to reflect the authentic performance of the experiments and will be less likely to disturb our analysis as those in the results of $C_{PT}$.

| Eval. Model | Method | R-Precision($\uparrow$) | | | FID ($\downarrow$) | MM-dist ($\downarrow$) | MR-Precision ($\uparrow$) | | |
|---|---|---|---|---|---|---|---|---|---|
| | | top 1 | top 3 | top 5 | | | top 1 | top 3 | top 5 |
| $C_{PT}$ | Real Motion | $0.193^{\pm.006}$ | $0.299^{\pm.002}$ | $0.413^{\pm.005}$ | $0.075^{\pm.066}$ | $5.151^{\pm.033}$ | - | - | - |
| | PT ($J_{PT}$) | $0.035^{\pm.009}$ | $0.082^{\pm.005}$ | $0.195^{\pm.004}$ | $4.855^{\pm.062}$ | $7.977^{\pm.023}$ | $0.088^{\pm.012}$ | $0.145^{\pm.012}$ | $0.212^{\pm.019}$ |
| | PT ($J_{ours}$) | $0.078^{\pm.004}$ | $0.149^{\pm.002}$ | $0.267^{\pm.003}$ | $5.135^{\pm.024}$ | $8.135^{\pm.019}$ | $0.138^{\pm.009}$ | $0.195^{\pm.023}$ | $0.311^{\pm.011}$ |
| | Ours | $0.165^{\pm.006}$ | $0.275^{\pm.009}$ | $0.356^{\pm.003}$ | $4.194^{\pm.037}$ | $4.899^{\pm.029}$ | $0.219^{\pm.017}$ | $0.325^{\pm.015}$ | $0.443^{\pm.056}$ |
| $C_{ours}$ | Real Motion | $0.425^{\pm.004}$ | $0.635^{\pm.006}$ | $0.733^{\pm.009}$ | $0.015^{\pm.059}$ | $2.413^{\pm.051}$ | - | - | - |
| | PT ($J_{PT}$) | $0.095^{\pm.004}$ | $0.155^{\pm.003}$ | $0.286^{\pm.002}$ | $3.561^{\pm.035}$ | $4.565^{\pm.027}$ | $0.175^{\pm.002}$ | $0.301^{\pm.010}$ | $0.419^{\pm.034}$ |
| | PT ($J_{ours}$) | $0.134^{\pm.002}$ | $0.285^{\pm.003}$ | $0.395^{\pm.005}$ | $3.157^{\pm.021}$ | $3.977^{\pm.024}$ | $0.216^{\pm.005}$ | $0.363^{\pm.006}$ | $0.489^{\pm.002}$ |
| | Ours | $\mathbf{0.389}^{\pm.006}$ | $\mathbf{0.575}^{\pm.009}$ | $\mathbf{0.692}^{\pm.005}$ | $\mathbf{1.335}^{\pm.003}$ | $\mathbf{2.856}^{\pm.009}$ | $\mathbf{0.497}^{\pm.006}$ | $\mathbf{0.691}^{\pm.004}$ | $\mathbf{0.753}^{\pm.015}$ |

Table 7: Quantitative comparison on Phoenix-2014 dataset, where **Real Motion** and **Ours** are evaluated by extracting the 3D keypoints from our mesh representation. The $J_{PT}$ and $J_{ours}$ in the bracket represent being trained on the corresponding keypoints.

Furthermore, we also qualitative comparison results in 13. Please see more visualizations in our supplementary video, and project page.

**Discussion**. With SignAvatars, our goal is to provide an up-to-date, publicly available 3D holistic mesh **motion-based** SLP benchmark and we invite the community to participate. As an alternative for the re-evaluation, we can also develop a brand new 3D sign language translation (SLT) method to **re**-evaluate PT and compare it with our method on BLEU and ROUGE. As a part of our future work on SL understanding, we also encourage the SL community to develop back-translation and mesh-based SLT methods trained with our benchmark. We believe that the 3D holistic mesh representation presents significant improvements for the accurate SL-motion correlation understanding, compared to the pure 2D methods as shown in Tab. 4 and Tab. 5, which was also proved to be true in a latest 3D SLT work (Lee et al., 2023).

## C    IMPLEMENTATION DETAILS FOR EXPERIMENTS AND EVALUATION

**Optimization strategy of automatic annotation pipeline**. During optimization, we utilize an iterative five-stage fitting procedure to minimize the objective function and use Adam optimizer with 1e-2 as the learning rate. Moreover, a good initialization can significantly boost the fitting speed of our annotation pipeline. At the same time, a well-pixel-aligned body pose will also help the reconstruction of hand meshes. Motivated by this, we apply 2000 fitting steps for a clip and split the fitting steps into five stages with 400 steps in each stage to formulate our iterative fitting pipeline. In the meantime, the Limited-memory BFGS (Nocedal & Wright, 1999) with a strong Wolfe line is applied to our optimization. In the first three stages, all the loss and parameters are optimized together. The weights $w_{body} = w_{hand} = 1$ are applied for $L_J$ to obtain a good body pose estimation. In the last two stages, we will first extract a mean pose from the record of the previous optimization to gain a stable body shape and freeze it as a fixed shape, as the signer will not change in a video by default. Subsequently, to obtain accurate and detailed hand meshes, we will enlarge the $w_{hand}$ to 2 to reach the final holistic mesh reconstruction with a natural and accurate hand pose.

### C.1    EVALUATION PROTOCOLS

In this subsection, we will elaborate on the computational details of our used evaluation protocol. To start with, our evaluation relies on a text-motion embedding model following prior arts (Zhang et al., 2023b; Tevet et al., 2022; Lee et al., 2023). For simplicity, we use the same symbols and notations as in our Secs. 3.1 and 4. Through the GRU embedding layer, we embed our motion representation $M_{1:T}$ and linguistic feature $E_{1:s}^l$ into $f_m \in R^d$ and $f_l \in R^d$ with the same dimensions to apply

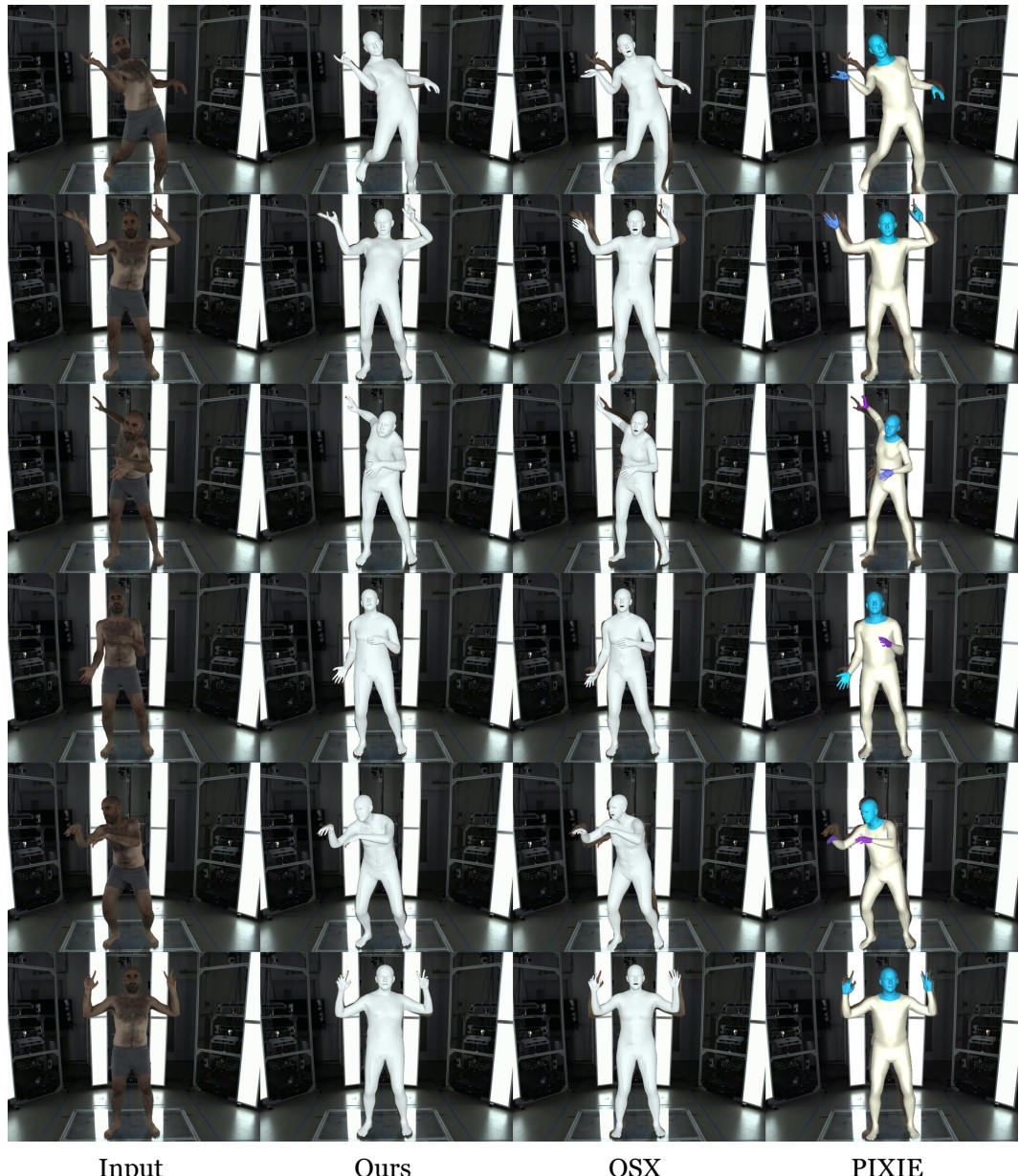

|            |            |            |            |
| :--------: | :--------: | :--------: | :--------: |
| Input      | Ours       | OSX        | PIXIE      |

Figure 10: Comparisons of existing 3D holistic human mesh reconstruction methods on EHF dataset. Our annotation method produces significantly better holistic reconstructions with plausible poses, as well as the best pixel alignment. (Zoom in for a better view)

contrastive loss and minimize the feature distances, where $d = 512$ is used in our experiments. After motion and prompt feature extraction, we compute each of the evaluation metrics, which are summarized below:

- **Frechet Inception Distance (FID)** ($\downarrow$), the distributional distance between the generated motion and the corresponding real motion based on the extracted motion feature.
- **Diversity**, the average Euclidean distance in between the motion features of $N_D = 300$ randomly sampled motion pairs.
- **R-precision** ($\uparrow$), the average accuracy at top-$k$ positions of sorted Euclidean distances between the motion embedding and each GT prompt embedding.

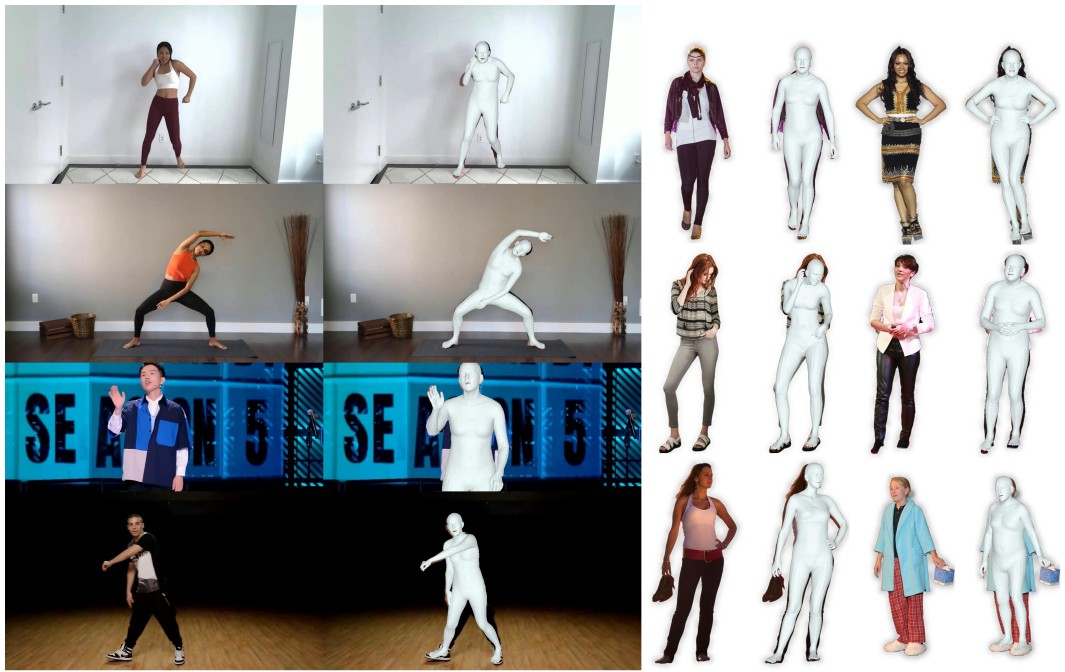

Figure 11: Our 3D holistic human mesh reconstruction methods on in-the-wild cases. (Zoom in for a better view)

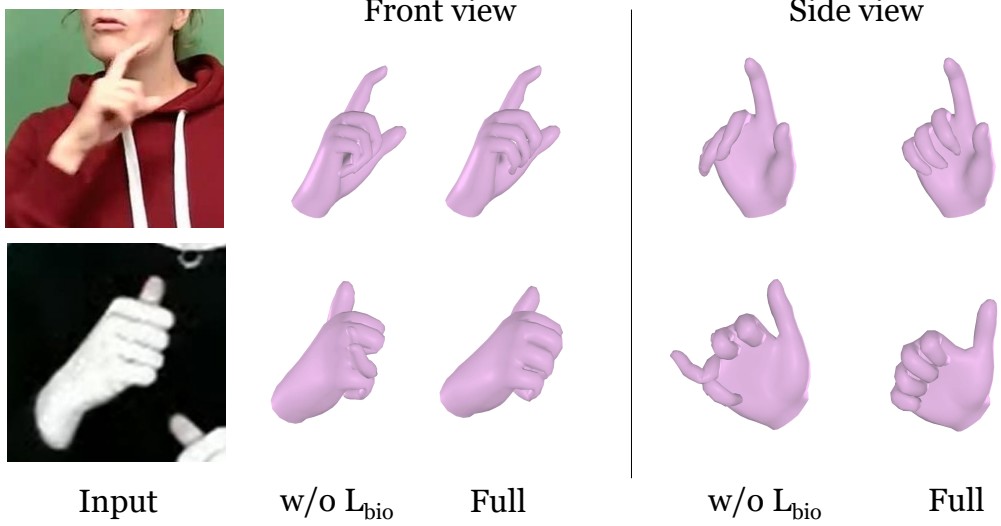

Figure 12: Visualization examples and analysis of our regularization term. The biomechanical constraints can alleviate the implausible poses caused by monocular depth ambiguity, which happens occasionally in complex interacting-hands scenarios for other monocular capture methods.

- **Multimodality**, average Euclidean distance between the motion feature of $N_m = 10$ pairs of motion generated with the same single input prompt.
- **Multimodal Distance (MM-Dist)** ($\downarrow$), average Euclidean distance between each generated motion feature and its input prompt feature.
- **MR-precision** ($\downarrow$), the average accuracy at top-$k$ positions of sorted Euclidean distance between a generated motion feature and 16 motion samples from dataset (1 positive + 15 negative).

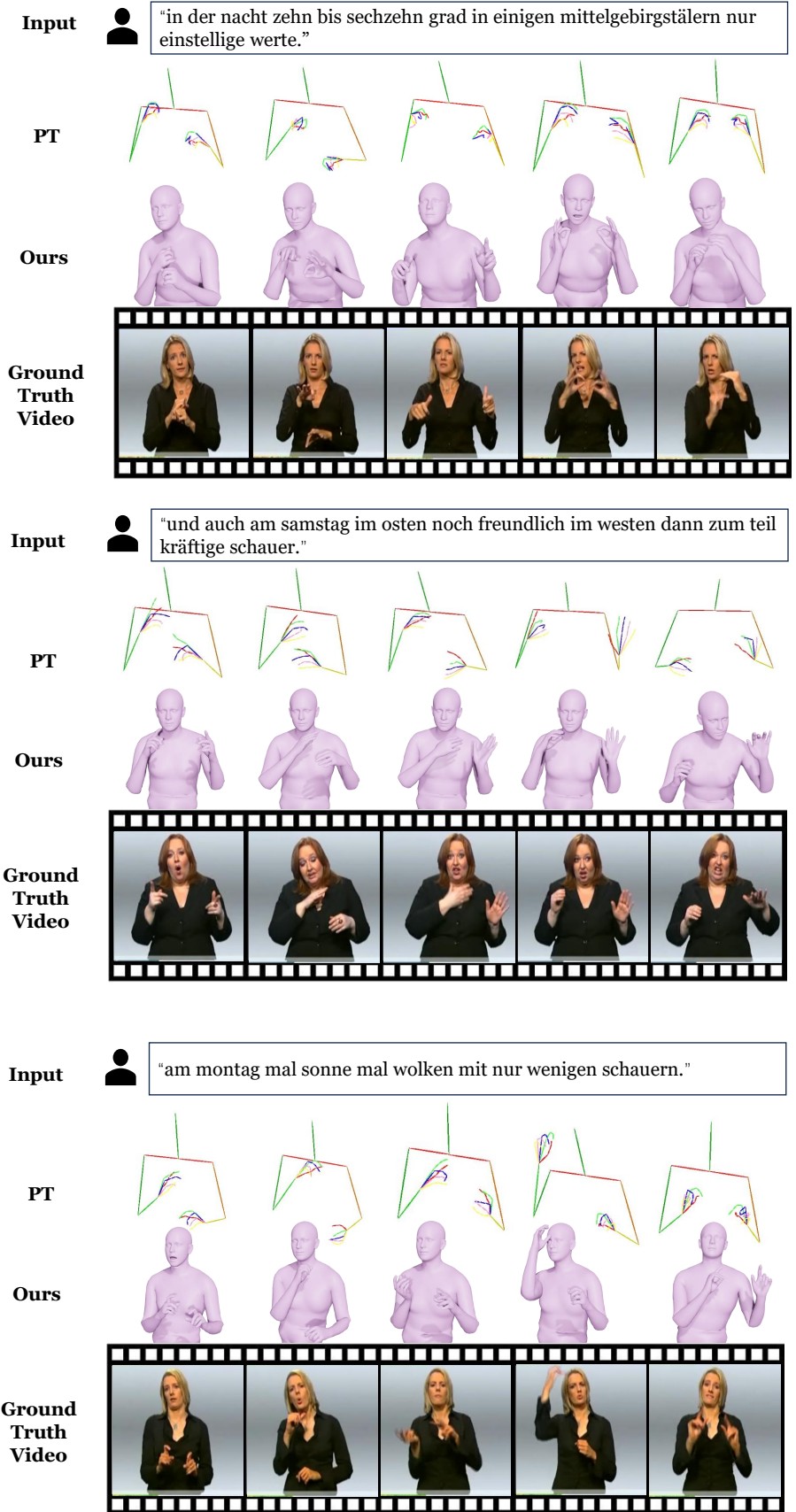

Figure 13: Qualitative comparison with PT (Saunders et al., 2020b) on Phoenix-2014 T dataset.

We now provide further details in each of those. For simplicity, we denote the dataset length as $N$ below.

**Frechet Inception Distance (FID).** is used to evaluate the distribution distance between the generated motion and the corresponding real motion:

$$FID = ||\mu_{gt} - \mu_{pred}||_2 - Tr(C_{gt} + C_{pred} - 2(C_{gt}C_{pred})^{1/2}) \tag{7}$$

where $\mu_{gt}, \mu_{pred}$ are the mean values for the features of real motion and generated motion, separately. $C, Tr$ are the covariance matrix and trace of a matrix.

**Diversity** is used for evaluating the variance of the generated SL motion. Specifically, we randomly sample $N_D = 300$ motion feature pairs $\{f_m, f_m'\}$ and compute the average Euclidean distance between them by:

$$Diversity = \frac{1}{N_D} \sum_{i}^{N_D} ||f_m^i - f_m^{i'}|| \tag{8}$$

**Multimodality** is leveraged to measure the diversity of the SL motion generated from the same prompts. Specifically, we compute the average Euclidean distance between the extracted motion feature of $N_m = 10$ pairs $\{f_m^j, f_m^{j'}\}$ of motion generated with the same single input prompt. Through the full dataset, it can be written as:

$$Multimodality = \frac{1}{N * N_m} \sum_{i}^{N} \sum_{j}^{N_M} ||f_m^{ij} - f_m^{ij'}|| \tag{9}$$

**Multimodal Distance (MM-Dist)** is applied to evaluate the text-motion correspondency. Specifically, it computes the average Euclidean distance between each generated motion feature and its input prompt feature:

$$MM\text{-}Dist = \frac{1}{N} \sum_{i}^{N} ||f_m^i - f_l^i|| \tag{10}$$

## D DISCUSSION

### D.1 RELATED WORK

In this section, we present more details about the related work as well as the open problems.

**Background**. Existing SL datasets, and dictionaries are typically limited to 2D, which is ambiguous and insufficient for learners as introduced in Lee et al. (2023), different signs could appear to be the same in 2D domain due to depth ambiguity. In that, 3D avatars and dictionaries are highly desired for efficient learning (Naert et al., 2020), teaching, and many downstream tasks. However, The creation of 3D avatar annotation for SL is a labor-intensive, entirely manual process conducted by SL experts and the results are often unnatural (Aliwy & Ahmed, 2021). As a result, there is not a unified large-scale multi-prompt 3D sign language holistic motion dataset with precise hand mesh annotations. The lack of such 3D avatar data is a huge barrier to bringing these meaningful applications to Deaf community, such as 3D sign language production (SLP), 3D sign language recognition (SLR), and many downstream tasks such as digital simultaneous translators between spoken language and sign language in VR/AR.

**Open problems**. Overall, the open problems chain is: **1)** Current 3D avatar annotation methods for sign language are mostly done manually by SL experts and are labor-intensive. **2)** Lack of generic automatic 3D expressive avatar annotation methods with detailed hand pose. **3)** Due to the lack of a generic annotation method, there is also a lack of a unified large-scale multi-prompt 3D co-articulated continuous sign language holistic motion dataset with precise hand mesh annotations. **4)** Due to the above constraints, it is difficult to extend sign language applications to highly desired 3D properties such as 3D SLR, 3D SLP, which can be used for many downstream applications like virtual simultaneous SL translators, 3D dictionaries, etc.

According to the problem chain, we will introduce the SOTA from three aspects: 3D holistic mesh annotation pipeline, 3D sign language motion dataset, and 3D SL applications.

**3D holistic mesh annotation:** There are a lot of prior works for reconstructing holistic human body from RGB images with parametric models like SMPL-X (Pavlakos et al., 2019), Adam (Joo et al., 2018). Among them, TalkSHOW (Yi et al., 2023) proposes a fitting pipeline based on SMPLify-X (Pavlakos et al., 2019) with a photometric loss for facial details. OSX (Lin et al., 2023b) proposes a time-consuming finetune-based weakly supervision pipeline to generate pseudo-3D holistic annotations. However, such expressive parametric models have rarely been applied to the SL domain. Kratimenos et al. (2021) use off-the-shelf methods to estimate holistic 3D mesh on the GSLL sign-language dataset (Theodorakis et al., 2014). In addition to that, only a concurrent work (Forte et al., 2023) can reconstruct 3D holistic mesh annotation using linguistic priors with group labels obtained from a sign-classifier trained on Corpus-based Dictionary of Polish Sign Language (CDPSL) (Linde-Usiekniewicz et al.), which is annotated with HamNoSys As such, it utilizes an existing sentence segmentation methods (Renz et al., 2021) to generalize to multiple-sign videos. These methods cannot deal with the challenging self-occlusion, hand–hand and hand–body interactions which makes them insufficient for complex interacting hand scenarios such as sign language. There is not yet a generic annotation pipeline that is sufficient to deal with complex interacting hand cases in **continuous and co-articulated** SL videos.

**Sign Language Datasets** While there have been many well-organized continuous SL motion datasets (Duarte et al., 2021; Albanie et al., 2021; 2020; Camgoz et al., 2018; Hanke et al., 2020; Huang et al., 2018) with 2D videos or 2D keypoints annotations, the only existing 3D SL motion dataset with 3D holistic mesh annotation is in (Forte et al., 2023), which is purely **isolated sign based** and not sufficient for tackling real-world applications in natural language scenarios. There is not yet a unified large-scale **multi-prompt 3D** SL holistic motion dataset with **continuous and co-articulated** signs and precise hand mesh annotations.

**SL applications** Regarding the SL applications, especially sign language production (SLP), Arkushin et al. (2023) can generate 2D motion sequences from HamNoSys. Saunders et al. (2020b) and Saunders et al. (2021b) are able to generate 3D keypoint sequences with glosses. The avatar approaches are often hand-crafted and produce robotic and unnatural movements. Apart from them, there are also early avatar approaches (Ebling & Glauert, 2016; Efthimiou et al., 2010; Bangham et al., 2000; Zwitserlood et al., 2004; Gibet et al., 2016) with a pre-defined protocol and character.

## D.2 LICENSING

Our dataset will first be released under the CC BY-NC-SA (Attribution-NonCommercial-Share-Alike) license for research purposes. Specifically, we will release the SMPL-X/MANO annotation and provide the instruction to extract the data instead of distributing the raw videos. We also elaborate on the license of the data source we used in our dataset collection:

**How2Sign (Duarte et al., 2021)**. Creative Commons Attribution-NonCommercial 4.0 International License.

**DGS Corpus (Prillwitz et al., 2008)**. is under CC BY-NC license.

**Dicta-Sign**. is under CC-BY-NC-ND 4.0 license.

**WLASL (Li et al., 2020)**. Computational Use of Data Agreement (C-UDA-1.0)

