# OpenReview forum: "SignAvatars: A Large-scale 3D Sign Language Holistic Motion Dataset and Benchmark"
_ICLR.cc/2024/Conference — Submitted to ICLR 2024_

### Official Review · Reviewer_MwWu · 2023-10-27

**Soundness:** 4 excellent
**Presentation:** 4 excellent
**Contribution:** 4 excellent
**Rating:** 8
**Confidence:** 4

**Summary:**

This paper looks at the problems of Sign Language (SL) production and translation. They introduce a significant new dataset of annotated sign language data. While many of the videos used are curated from other datasets (e.g., How2Sign), a core contribution is their optimized 3D body (SMPL-X) and hand (MANO) representations for each frame in these datasets. Sign Language transcription forms can vary (e.g., text/gloss/HamNoSys), so one benefit of this work is the inclusion of multiple forms and across languages (English, German SL, French SL, Polish SL, Greek SL). The authors also introduce benchmarks and metrics to facilitate future modeling work.

On top of this, the authors describe development of an SL production baseline based on VQ-VAE models which shows promise. The videos are compelling and the results are significantly better than Ham2Pose on various metrics.

**Strengths:**

This is an exceptional paper that will be important for the sign language modeling community from dataset, benchmarking, and modeling perspectives.

Some things that I found are really well done / investigated.
* The human pose representations build on SOTA avatar representations (SMPL+X, MANO) as opposed to the more common keypoint-based solutions.
* For computing human pose annotations, the authors do an especially nice job with novel system engineering and iteration to compute the highest quality annotations. For example, adding biomechanics constraints on top of the SMPL and MANO computations.
* The use of multiple SL annotation types (HymNoSys, text, gloss) makes this work useful for working on SL modeling from multiple perspectives.
* The SL production work is a really nice advanced development of discrete variable model approaches for complex motion synthesis.
* Related work is well represented and contextualized (although some additional work from the HCI and Accessibility communities could be added)
* The paper is well written and generally easy to follow (but may be hard to reproduce given the complexities of each model involved).

**Weaknesses:**

* The number of metrics is a little overwhelming. Some of them seem very useful but others may dilute the findings? For example, I'm not convinced that metrics like FID are useful here. Can the authors demonstrate that improved results on each metric do correlate with a model's ability to generate correct/accurate signs?

**Questions:**

Perhaps I missed it, but how important are the biomechanics constraints or other regularizations on the annotation quality? Are there metrics for this or is it mostly perceptual.

---

> ### Author Response · Authors · 2023-11-22
> **Author Response**
>
> We thank the reviewer for their considerate feedback. We appreciate that they recognize the impact and significance of the SMPL-X and MANO annotations in our dataset as well as its multi-prompt and cross-language nature. We are delighted that the reviewer finds our system design novel and paper exceptional. We confirm that the reviewer has a good grasp of the contributions of our paper.
>
> We agree with the reviewer that some of the de-facto metrics used ubiquitously do not reflect the performance very well for the task at hand. $\textbf{FID}$ could be one of them. However, rather than omitting these metrics, we find it important to have them available as they are used across a wide range of works and make cross-checking easier. Nonetheless, we only employ FID to evaluate the first stage of VQVAE training. For this part, FID provides hints on whether our VQVAE is generating reasonable and plausible motions with a similar distribution to our ground-truth annotation.
>
> New videos in $\textbf{our project webpage (see general remarks)}$, correlate well with our low-FID scores and demonstrte the realism and stability of our motion generation. Following reviewer's suggestion, in the project webpage we included character-driven animation to produce more visible motion patterns. In addition, we have $\textbf{enriched our Appendix}$ with Figures $\textcolor{red}{10-13}$, where the benefits are qualitatively shown.
>
> Evaluating the impact of biomechanical constraints is indeed a good idea. To this end, we have included a version of our method ablating the biomechanical constratints in Table $\textcolor{red}{6}$ of our $\textbf{revised Appendix}$. This table can also be found in response to Reviewer qy3Y. Furthermore, Figure $\textcolor{red}{12}$ qualitatively depicts the the impact of these constraints, where the reconstructed poses are essentially more realistic and plausible.

---

> > ### Comment · Reviewer_MwWu · 2023-11-23
> > **Keeping scores**
> >
> > I am confirming that I have read the other reviews and rebuttals. I am keeping my score ("accept").

---

> ### Author Response · Authors · 2023-11-23
> **Author Response**
>
> We thank the reviewer for the recognition, especially regarding the impact and significance of our work. Your precious comments are indeed encouraging. Please feel free to follow up on the updates of our process, evaluation analysis, and applications on our anonymous GitHub page.
>
> All the best,

---

### Official Review · Reviewer_Dj8e · 2023-10-30

**Soundness:** 3 good
**Presentation:** 3 good
**Contribution:** 2 fair
**Rating:** 5
**Confidence:** 5

**Summary:**

In this paper, the authors apply a parametric body model based 3D pose estimation framework to estimate signer poses from monocular videos. They apply their framework on publicly available SL datasets. The extracted poses are noted to be publicly available upon publication of the manuscript to set up a new benchmark for SLP. In addition to the avatar generation framework, the authors propose an VQ-VAE based SLP approach, which can be prompted by spoken language sentences, words, sign glosses or HamNoSys sequences. This approach is evaluated on the presented benchmark, and the presented qualitative results look promising.

**Strengths:**

- It's an interesting idea to have multi-source prompting as SLP input.

- The qualitative samples the authors share seem to yield compelling SLP performance.

- Although VQ-VEAs have been explored for SLP (https://arxiv.org/abs/2208.09141), it has not been utilized in combination with multi-source prompting and 3D mesh generation to the best of my knowledge. Hence, this approach and the results might be useful to the other researchers.

**Weaknesses:**

General:
- Supplementary video could have been better. There is no narration in the video. Failure analysis and having SLP with different prompts that are corresponding to the same meaning would have strengthened the submission.

- Overall the presentation quality of the paper can improve significantly. As is it does not meet the expectation of being publication ready (See Questions).

About Dataset:
- This manuscript is framed as a dataset paper, however there is no new data that is collected or will be released. What the paper actually presents is derivative data, i.e. 3D pose estimates from existing datasets, which is disappointing as a reader who was hoping to find a new,  potentially useful data source.

- Although the authors present this dataset as "large scale", it still lacks the scale to be considered as one. 70,000 videos is hardly large scale, even considering the contemporary SL alternatives, such as BOBSL or Youtube-ASL.

About Pose Estimation Framework:
- Given one of the main proposed contributions of this paper is the pose estimation framework, I would have expected more qualitative and quantitative results against the state-of-the-art approaches from the model based pose estimation domain. If this is just an application of previously existing approaches, such as (Spurr et al., 2020), then this needs to be clearly stated.

About VQ-VAE SLP-based Approach:
- The proposed VQ-VAE based SLP approach is only evaluated on the presented benchmark dataset, which does not give the reader any anchor points to compare against the state-of-the-art on other benchmarks. Also the authors compare their approach only against the Ham2Pose-3D approach on the new benchmark.

**Questions:**

- "We compile SignAvatars by synergizing various data sources from public datasets to online videos and form seven
subsets, whose distribution is reported in Fig. 2". - As it was used multiple times in the manuscript, what does "synergizing" mean in the context of this paper?

- Figure 2 is not clear. What is "word", which is the "ASL" dataset? It would have been better to have a clear introduction of the terminology and the source dataset that are utilized just after or before Figure 2.

- "Moreover, there are over 300 different sign languages across the world, with hearing-impaired people who do not know any SL." Can the authors elaborate what they mean here?

- "Our SL annotations can be categorized into four common types: HamNoSys, spoken language, word, and gloss, which can be used for a variety of downstream applications such as SLP and SLR". What is the difference between word and gloss in this context?

- "Overall, we provide 117 hours of 70K video clips with 8.34M frames of motion data with accurate expressive holistic 3D mesh as motion annotations" - Will the authors provide the original video clips? Did you ask the original authors permission? As you shared in your appendix not all the datasets have "Share alike" permission in their licenses.

- "To demonstrate our dedication, we have submitted all source code as part of the appendix." - I was unable to find the source code either in the appendix or the supplementary material.

- "RELATED WORK (REUSING THE PREVIOUS ANSWERS...)"- What do the authors mean by "Reusing the previous answers"?

----------------------
**After Rebuttal Comments:**

Authors provided a great rebuttal and updated their website, which addressed most of my concerns, and I thank them for that.

I am still adamant about my standing of this not being a dataset paper, as I categorically do not consider automatic pose estimations as annotations.

There is a lot in this paper, and I do think it will be useful to the field, but as is, it is spread thin over many components of a sign language production pipeline. As I mentioned in my initial review, if the novelty comes from the pose estimations, then we need more experiments and comparisons against other pose estimation methods. If the novelty is the SLP network, then we need more comparisons against sota and a user study to understand its performance. Thankfully, the authors provided additional experiments in the discussion period, so some of these concerns are partially addressed.

Given all these, I am learning towards improving my recommendation to a borderline rating, and I do not feel strongly for either acceptance or rejection of this manuscript.

**Details Of Ethics Concerns:**

- Please use the phrase "Deaf and Hard of Hearing" instead of "hearing impaired" or "people suffering from hearing disability", as the latter is considered not appropriate and even offensive by the DHH communities.

- As mentioned in my questions, it is not clear to me if the authors are planning to share the videos from the original datasets the authors have utilized to compile their benchmark. Hence possible licensing issues may arise if the appropriate permissions have not been taken.

---

> ### Author Response · Authors · 2023-11-22
> **Author Response**
>
> We thank the reviewer for your detailed comments and careful perusal. We appreciate that the reviewer acknowledges the usefulness of our method and finds our SLP performance to be compelling, and multi-source prompting to be interesting especially when combined with 3D mesh generation.
>
> Importantly, we appreciate their warning on the use of terminology. We have now updated the manuscript to use either $\textbf{``Deaf" or `Hard of Hearing' everywhere}$.
>
> $\textbf{General.}$ We thank the reviewer for the suggestions. Our new supplementary video on our project page (see general comments) now contains narration. Rapid motions are still difficult for us to handle. Handling ambiguities in the SL videos is in fact an important future work.
>
> $\textbf{About Dataset.}$ We kindly disagree with the reviewer. There are several components to a dataset, two of which are the input and the labels or annotations. While we source our videos from various online repositories, we do compute our annotations (crucial for supervised learning) which indicates a new dataset. We will in fact be releasing these annotations upon publication.
>
> We agree with the reviewer that "large scale" is an ambiguous term used in reference to varying sizes for varying problems. Note that, we are a "3D motion" dataset and not a 2D video dataset.
> For example, a subset of our data, How2Sign, is already claimed to be large-scale for this setting. Nevertheless, our dataset is the largest as far as 3D meshes, SMPL-X and MANO annotations are concerned, making it a significant contribution in this field as confirmed by Reviewer MwWu.
>
> $\textbf{About Pose Estimation Framework.}$ We thank the reviewer for bringing this up. Note that our annotation pipeline (pose estimation or reconstruction algorithm) with its temporal-consistency, biomechanical constraints and hierarchical initialization is novel. To demonstrate its advantage, we have now included more qualitative and quantitative results, both regarding our dataset and EHF dataset [1]. Specifically, $\textbf{Appendix B.1}$ provides new evaluations of our reconstruction in comparison to the state-of-the-art. In the existing benchmark of EHF, our annotation method shows a clear advantage (see Table 6, which we also include in response to Reviewer qy3Y). We further illustrate this qualitatively in Figure $\textcolor{red}{10}$.
>
> [1] Pavlakos, Georgios, et al. "Expressive body capture: 3d hands, face, and body from a single image." CVPR 2019.

---

> > ### Author Response · Authors · 2023-11-22
> > **Author Response 2**
> >
> > $\textbf{VQ-VAE SLP.}$ Firstly, it is not correct that we compare only with Ham2Pose on our new benchmark. We have also compared with the diffusion model based on MDM [2], which is ``SignDiffuse'' in our Table. 5 of our main paper. We stated this clearer now.
> >
> > Next, the reviewer is right that we were only able to evaluate our SignVAE on our dataset. However, to the best of our knowledge, no publicly available benchmark for $\textbf{3D mesh and motion-based}$ SLP exists. As an alternative, Progressive Transformer [3] and its successors provide a $\textbf{keypoint-based}$ 3D Text2Pose (Language2Motion) benchmark based on PHOENIX2014T dataset [4] where back-translation derived metrics are used. Note, conducting back-translation evaluations as in must strictly use a common back-translation model checkpoint for a fair comparison. Unfortunately, the pretrained evaluation model checkpoint or its reproductions are available neither on the project website (https://github.com/BenSaunders27/ProgressiveTransformersSLP) or on other sites.
> >
> > To address the reviewer's concern, we have re-evaluated these benchmark methods as follows:
> > * First, we generate mesh annotations for the Phoenix-2014T dataset and add them as our subsets GSL. We follow the original data distribution and official split to train our network.
> > * Second, because in addition to the absence of the evaluation model, the generation model checkpoints are also lacking, we re-train PT using the official implementation on both 3D-lifted OpenPose keypoints $J_{PT}$ and the 3D keypoints $J_{ours}$ regressed from our mesh representation, corresponding to PT ($J_{PT}$) and PT ($J_{ours}$).
> > * Third, we train two 3D keypoints-based SL motion evaluation models on this subset with $J_{PT}$ and $J_{ours}$, resulting in two model checkpoints $C_{PT}$ and $C_{ours}$.
> >
> > Our results, presented in Appendix $\textcolor{red}{\text{B}.2}$ (Figure $\textcolor{red}{13}$ and Table $\textcolor{red}{7}$ below) show that our SignVAE surpasses the existing state-of-the-art methods by a large margin.
> >
> >
> > | **Eval. Model**   | **Method**       |  |            **R-Precision$\uparrow$**             |                          | **FID $\downarrow$** | **MM-dist $\downarrow$** |  |            **MR-Precision $\uparrow$**              |                          |
> > | :---------------- | :--------------- | :----------------------------: | :----------------------: | :----------------------: | :-----------------------: | :-------------------------------: | :------------------------------: | :----------------------: | :----------------------: |
> > |                   |                  | top 1                          | top 3                    | top 5                    |                           |                                   | top 1                            | top 3                    | top 5                    |
> > | **C$_{PT}$**   | **Real Motion**  | 0\.193           | 0\.299     | 0\.413     | 0\.075      | 5\.151              | -                                | -                        | -                        |
> > |                   | PT (J$_{PT}$)  | 0\.035           | 0\.082     | 0\.195     | 4\.855     | 7\.977              | 0\.088             | 0\.145     | 0\.212     |
> > |                   | PT (J$_{Ours}$) | 0\.078           | 0\.149    | 0\.267     | 5\.135      | 8\.135              | 0\.138             | 0\.195     | 0\.311     |
> > |                   | Ours             | 0\.165           | 0\.275     | 0\.356     | 4\.194      | 4\.899              | 0\.219             | 0\.325     | 0\.443     |
> > | **C$_{ours}$** | **Real Motion**  | 0\.425           | 0\.635     | 0\.733     | 0\.015      | 2\.413              | -                                | -                        | -                        |
> > |                   | PT (J$_{PT}$)   | 0\.095           | 0\.155     | 0\.286     | 3\.561      | 4\.565              | 0\.175             | 0\.301     | 0\.419     |
> > |                   | PT (J$_{ours}$) | 0\.134          | 0\.285     | 0\.395     | 3\.157      | 3\.977              | 0\.216             | 0\.363     | 0\.489     |
> > |                   | Ours             | **0\.389**       | **0\.575** | **0\.692** | **1\.335**  | **2\.856**          | **0\.497**        | **0\.691** | **0\.753** |
> >
> > We refer the reviewer to our $\textbf{revised Appendix}$ for further details.
> >
> > $\textbf{References}$
> >
> > [2] Tevet, Guy, et al. "Human Motion Diffusion Model." ICLR 2022.
> >
> > [3] Saunders, Ben, et al. "Progressive transformers for end-to-end sign language production." ECCV 2020.
> >
> > [4] Camgoz, Necati Cihan, et al. "Neural sign language translation." CVPR 2018.

---

> > > ### Author Response · Authors · 2023-11-22
> > > **Author Response 3**
> > >
> > > $\textbf{Questions.}$ We have fixed the typos in our revision and marked them in blue. We thank the reviewer for this.
> > >
> > > * $\textbf{Synergizing}.$ We use the term "synergize" in the context where we source multiple datasets from different repositories and domains yielding a whole that is more effective than the sum of its parts. We now rephrased this as "gathering" or "combining" to aid clarity.
> > >
> > > * $\textbf{word / ASL dataset.}$ The source datasets were introduced in the "Data source" part of Section 3.1 and the source data distribution is actually presented in Table 2, instead of Figure 2. We corrected this typo in the caption of Table 2 by stating $\textbf{``source datasets'' instead of `subsets'}$. For example, the ASL (American Sign Language) and GSL (German Sign Language) datasets are co-articulated but annotated in different sign languages (American and German). The other two groups, HamNoSys and Word subsets, are isolated having HamNoSys and single-word annotations, respectively. We now clarified this.
> > >
> > > * $\textbf{``300 SL''.}$ Despite the existence of more than 300 sign languages, there are still Deaf communities in the world who donot possess the knowledge of any of these sign languages.
> > >
> > > * $\textbf{Word vs. GLOSS.}$ Technically, GLOSS is a label for the sign to easy talking about it. It is written all upper case to distinguish from words. It is not the translation and does not convey the sign's meaning itself. Moreover, different words might be written as the same gloss (e.g. swim, swan, swum all correspond to 'SWIM'). Regarding the usage in our work, the "gloss" specifically refers to the continuous and sentence-level gloss annotation to distinguish them from other single signs videos with only HamNoSys or isolated word-level annotation. The "word" specifically refers to the abbreviation of "word-level annotation", the samples with isolated signs and single word/gloss annotation. We have changed "gloss" and its abbreviation "G" to "sentence-level gloss" and "SG", and added more details and descriptions in Section 3.1 to make it clearer.
> > >
> > > * $\textbf{Providing original video clips.}$ We thank the reviewer for mentioning this. We will strictly follow the licenses of the source videos and will not distribute the original video clips under the current license. We will provide our SMPL-X/MANO parameters as mesh annotations and provide detailed instructions for motion extraction. We have included licensing notes in our Appendix.
> > >
> > > * $\textbf{Data and code release.}$ Our dataset and code will be released soon upon acceptance.
> > >
> > > We thank the reviewers for bringing these up and have now improved the paper accordingly.

---

### Official Review · Reviewer_qy3Y · 2023-11-01

**Soundness:** 3 good
**Presentation:** 3 good
**Contribution:** 3 good
**Rating:** 6
**Confidence:** 2

**Summary:**

This paper presents a large-scale multi-cue 3D sign language (SL) action dataset, aiming to build a communication bridge for hearing-impaired individuals.

**Strengths:**

- Well-written.
- A practical dataset is proposed.

**Weaknesses:**

- Why did the author use the annotation method in Figure 3? Are there other labeling methods that can be compared?
- It is expected that the author can describe the specific structure of the "Autoregressive Transformer" in Figure 4.
- What is the specific meaning of "code index vector" in Figure 4? Please clarify.

**Questions:**

Please see "Weaknesses".

---

> ### Author Response · Authors · 2023-11-22
> **Author Response**
>
> We thank the reviewer for their feedback.
>
> $\textbf{1. Are there other labeling methods that can be compared?}$ We agree with the reviewer that other annotation methods can be used and thank the reviewer for bringing this up. We have considered this suggestion and have evaluated our method against other potential annotation strategies, on a dataset where precise ground truth is available, specifically the EHF dataset [1]. In particular, we choose the state-of-the-art methods of OSX [2], PyMafX [3], and Pixie [4], and present a comparison in $\textbf{Table 6}$ of our $\textbf{revised Appendix}$. We also include this table below:
>
> | **Method**                                          | **MPVPE** |       |       | **PA-MPVPE** |       |       | **PA-MPJPE** |       |
> | :---------------------------------------------------- | :--------: | :---: | :---: | :------------: | :---: | :---: | :--------------: | :---: |
> |                                                       | Holistic   | Hands | Face  | Holistic       | Hands | Face  | Body             | Hands |
> | SMPLify-X [1] *                        | -          | -     | -     | 65\.3          | 75\.4 | 12\.3 | 62\.6            | 12\.9 |
> | FrankMocap [5]   $\dagger$         | 107\.6     | 42\.8 | -     | 57\.5          | 12\.6 | -     | 62\.3            | 12\.9 |
> | PIXIE [4]  $\dagger$                       | 89\.2      | 42\.8 | 32\.7 | 55\.0          | 11\.1 | 4\.6  | 61\.5            | 11\.6 |
> | Hand4Whole [6] $\dagger$ | 76\.8      | 39\.8 | 26\.1 | 50\.3          | 10\.8 | 5\.8  | 60\.4            | 10\.8 |
> | PyMAF-X [3]    $\dagger$                   | 64\.9      | 29\.7 | 19\.7 | 50\.2          | 10\.2 | 5\.5  | 52\.8            | 10\.3 |
> | OSX [2]    $\dagger$                              | 70\.8      | -     | -     | 48\.7          | -     | -     | 55\.6            | -     |
> | Motion-X [7]   $\ddagger$               | 44\.7      | -     | -     | 31\.8          | -     | -     | 33\.5            | -     |
> | Motion-X w/GT 3Dkpt [7]   $\ddagger$   | 30\.7      | -     | -     | 19\.7          | -     | -     | 23\.9            | -     |
> | **Ours (w/o bio)** *                                 | 21\.6      | 12\.5 | 7\.8  | 14\.2          | 5\.4  | 4\.3  | 16\.5            | 6\.2  |
> | **Ours**  *                                         | **20\.1**      | **9\.7**  | **7\.8**  | **12\.9**           | **4\.7**  | **4\.3**  | **15\.6**            | **5\.8**  |
>
> We further present in Figures $\textcolor{red}{9}, \textcolor{red}{10}$ and $\textcolor{red}{11}$ of our $\textbf{revised Appendix}$, that our method provides significantly better quality regarding $\textbf{pixel alignment}$, with more natural and plausible hand poses. Please find more results in our updated videos on our anonymous project page, found in the general comments.
>
>
> $\textbf{2. Structure of autoregressive Transformer and code-index vector.}$
> Code index vector is the (discrete) latent feature of the motion. Let us consider Motion VAE in Figure 4. Any SL motions can be encoded into a continuous latent space, which can be discretized into a sequence of codebook indices via quantization using a learnable codebook. This is denoted as $X = [x_{1}, ..., x_{T/w}, x_{\mathrm{EOS}}]$, where $\mathrm{EOS}$ (end of sequence) is an extra learnable end token representing the stop signal. By projecting $X$ back to their corresponding codebook entries, we can obtain the $\textit{code index vectors}$ as the latent feature $F^m_{1:(T/w)}=(f_{1}^{m}, \dots , f_{1:(T/w)}^{m})$ and $f_{i}^{m} \in \mathbb{R}^{d_{z}}$.
>
> Autoregressive Transformer serves as an autoregressive code index generator and the objective for training the code index generator can be seen as an autoregressive next-index prediction task. The process can be formulated as follows: given the previous $i-1$ indices and the conditioning code $c$ embedded by fusing the linguistic feature $E^{l}$ and the codebook index vectors, the Transformer predicts the distribution of possible next indices $p(X_{i},c\mid X_{<i})$ as shown in Figure 4.
>
> $\textbf{References}$
>
> [1] Pavlakos, Georgios, et al. "Expressive body capture: 3d hands, face, and body from a single image." CVPR 2019.
>
> [2] Lin, Jing, et al. "One-Stage 3D Whole-Body Mesh Recovery with Component Aware Transformer." CVPR 2023.
>
> [3] Zhang, Hongwen, et al. "Pymaf-x: Towards well-aligned full-body model regression from monocular images." TPAMI 2023.
>
> [4] Feng, Yao, et al. "Collaborative regression of expressive bodies using moderation." 3DV 2021.
>
> [5] Rong, Yu, Takaaki Shiratori, and Hanbyul Joo. "Frankmocap: A monocular 3d whole-body pose estimation system via regression and integration." ICCV 2021.
>
> [6] Moon, Gyeongsik, Hongsuk Choi, and Kyoung Mu Lee. "Accurate 3D hand pose estimation for whole-body 3D human mesh estimation." CVPRW 2022.
>
> [7] Lin, Jing, et al. "Motion-x: A large-scale 3d expressive whole-body human motion dataset." NeurIPS 2023.

---

### Official Review · Reviewer_XQXu · 2023-11-02

**Soundness:** 2 fair
**Presentation:** 3 good
**Contribution:** 3 good
**Rating:** 6
**Confidence:** 5

**Summary:**

This paper proposes a large-scale 3D sign language motion dataset. This dataset is organized with video-mesh-prompt. For accurate mesh annotation, it designs multiple loss terms and leverages 2D pose detectors to provide supervision signal. For prompt, it collects multiple types, i.e., HamNoSys, spoken language, and words. Besides, it also provides a baseline for sign language production.

**Strengths:**

- To my best knowledge, this paper proposes the largest SL motion dataset. It promotes the research in sign language production.
- The designed baseline is sound.
- The whole paper is well-organized and well-written.

**Weaknesses:**

- One of the main concerns is the evaluation metrics. It is important to evaluate the semantics of the produced motion. Although the authors claim that the metric of back-translation is not generic for each text prompt, we can divide the dataset into multiple groups, i.e., word-level and sentence-level. Word-level and sentence-level videos should be divided, as they have different co-articulated characteristic.
- For the proposed baseline method, how does the semantics input act as a condition in the autoregressive Transformer?
- What is PLFG? I cannot find this module in Figure 4.
- The core design of the baseline is the utilization of VQ-VAE for both motion and semantics tokenization. Could the authors perform ablation on it?
- Some typos, divrse in Page 4; the ASL data volume is not consistent in Table 2 and the text description (34K, 35K).
- Some other relevant works should be discussed in the part of 3D holistic mesh reconstruction (for SL), such as
Hu H, Zhao W, Zhou W, et al. SignBERT+: Hand-model-aware Self-supervised Pre-training for Sign Language Understanding. IEEE TPAMI, 2023.

**Questions:**

Please refer to the weakness part.

---

> ### Author Response · Authors · 2023-11-21
> **Author Response**
>
> We thank the reviewer for their constructive feedback and appreciate that the reviewer acknowledges the impact of our SL motion dataset on sign language production as well as the soundness of our baseline and the quality of our manuscript. We have incorporated the minor corrections and additional citations. We address the rest of the concerns below.
>
> - $\textbf{Evaluation metrics.}$ We absolutely agree with the reviewer that, due to the novelty of the task and our benchmark, evaluation needs to be rethought. Unfortunately, we are not aware of a direct 3D back-translation model for motion SLP, which we plan to investigate in a future study. In this work, instead of re-inventing the wheel, we indeed contribute to the available repertoire of metrics. In particular, inspired by the studies in human-motion-generation, we introduce MR-precision, which evaluates how much $\textit{generated motion}$ resembles the $\textit{ground truth motion}$. As we describe in our Appendix C1, it measures the average accuracy at top-$k$ positions of sorted Euclidean distance between a motion feature generated from a sequence joint-rotations and 16 motion samples from dataset. We argue that, as opposed to mean 3D joint position error, this metric is especially suited to the 3D  motion SLP task since: (i) the orientation of hand joints carries particular semantic meaning, (ii) such comparison in the latent space parallels a $\textit{semantic evaluation}$ of signs, without having the need for an explicit continuous-mesh to sign classifier. In the future, we will work on designing more explicit, physical evaluation-models. We have now updated the paper for further clarity.
>
> - $\textbf{Baseline semantics input.}$ $\textbf{In our SignVAE}$, the semantics input to our method is the linguistic feature $E^{l}$. As in Figure 4, our conditioning code $c$ is formed by concatenating this semantics input and the codebook index vectors of $Z^{l}$, whose dimension is reduced into the transformer dimension by a linear projection. We subsequently concatenated this feature with the current $\textit{code index vector}$ for a positional embedding. $\textbf{In our baseline in Table 5}$, SignVAE (Base), we remove the PLFG, and replace the conditioning code $c$ with a canonical CLIP-style encoder feature with 512 dimensions.
>
> - $\textbf{PLFG}$ is our parallel linguistic feature generator, which is the VQVAE based structure illustrated at the top of Figure 4. It is a linguistic feature generator jointly optimized with our SL motion generator (bottom of Figure 4) in a parallel manner. We have polished the description of this part in our revised version.
>
> - $\textbf{Ablation on VQVAE.}$ We agree with the reviewer that ablations on our VQVAE are important. This is the reason why Table 5 of our main paper ablates on the essential components of our VQVAE, namely the parallel linguistic feature generator (PLFG) and the quantized semantic representation.
>
> - $\textbf{Comparison to SignBERT+ on reconstruction.}$ We have properly cited and discussed SignBERT+ in our revised paper.

---

### Public Comment · ~Maria-Paola_Forte1 · 2023-11-21
**Discrepancies in Description of SGNify**

The paper incorrectly describes SGNify, the method proposed by Forte et al. Specifically, SGNify does not need HamNoSys annotations to reconstruct sign-language videos. SGNify can automatically reconstruct sign-language videos without any prior knowledge about the HamNoSys notation, the sign in the video, or the specific sign language. Furthermore, Forte et al. also showed the extension of SGNify to sentences. I would be happy to explain more about how SGNify works; please feel free to reach out to me at forte@is.mpg.de.

---

> ### Author Response · Authors · 2023-11-22
>
> Dear Maria,
>
> We acknowledge the oversight in our paper regarding the use of HamNoSys. Specifically, while SGNfiy does not directly require HamNoSys, it does necessitate labels for training a group classifier. These labels are typically derived from HamNoSys annotations, as stated by Forte et al. in the parsing of the Corpus-based Dictionary of Polish Sign Language (CDPSL). Though, to the best of our understanding, the paper does not provide alternative methods for obtaining these labels beyond HamNoSys parsing. We have clarified this solely implicit dependence on HamNoSys in our revised manuscript.
>
> We now reflect the recent update that the SGNify 3D SL holistic dataset was released back in June, and mention that SGNify, enhanced by the methodologies of Renz et al., can process continuous signs, despite the dataset comprising only isolated signs (the primary focus of SGNify).
>
> We would kindly like to point out that our work does not distinguish between isolated or continuous videos, and treats continuous videos naturally within its pipeline. This seamless integration removes the necessity for external sentence-level segmentation, allowing us to bypass the computation of group labels. Moreover, the SignAvatars dataset includes both continuous and isolated sign language videos.
>
> It is important to stress that our SignVAE significantly departs from SGNify. We introduce a generative model for 3D sign language videos, enabling the exploration of new tasks such as 3D motion sign language production.
>
> We are committed to adhering to all reviewing guidelines and look forward to further communication with you post-review stage.
>
> All the best,

---

### Author Response · Authors · 2023-11-22
**General comment**

We thank the reviewers for their comprehensive comments and appreciation. We have now revised our paper according to the reviewers' comments, which we also respond below. We particularly verify the effectiveness of our annotation pipeline in the EHF dataset, as well as assess the SignVAE generation on Phoenix2014T. These quantitative evaluations as well as corresponding qualitative results are found in our $\text{revised Appendix}$ $\textbf{B}$. We will also discuss these in our comments below.

Our anonymized project page can be found at: https://anonymoususer4ai.github.io/.

---

### Meta-Review · Area_Chair_96Cj · 2023-12-05

**Metareview:**

Authors apply a parametric model-based 3D pose estimation framework to estimate signer poses ("avatars") from videos belonging to publicly available sign language datasets. Authors will make these annotations publicly available upon publication, to create a new "dataset and benchmark". In addition to the avatar generation framework, the authors propose a VQ-VAE based sign language processing approach, which can be prompted by either spoken language sentences, words, sign glosses or HamNoSys sequences (learning from multiple supervisions). This approach is evaluated on the presented benchmark, and the presented qualitative results look promising.

Strengths:
- It's an interesting idea to have multi-source prompting as SLP input, and novel (in combination VQ-VAE and 3D mesh)
- The qualitative samples the authors share seem to yield compelling sign language recognition performance, although results are only provided on the new annotations/ database
- The provided annotations will be useful to the community

Weaknesses:
- The paper has little focus. For ICLR, it may be preferable to focus on a novel method, e.g. training a single VQ-VAE model for different inputs (sign language, audio, glosses, etc) -- and provide more detailed ablations on these techniques (rather than on performance on different datasets)
- The motivation and presentation can be improved. Authors responded to reviewer feedback, but the term "deaf and hard of hearing" is still not used consistently throughout the paper
- The data size (70k samples) is not really "large scale" (although relatively large for the type of data). Unless the methods can be shown to perform well on other types of "low resource" data, this paper may best be targeted to a corresponding conference (Computer Vision, ACM Multimedia, or HCI, etc). ICLR audience would be more appropriate if the methods could be shown to generalize to other similar tasks.
- The novelty of the paper in the extraction algorithm seems to focus on sign language domain specifics (constraints, hierarchical structure) and may be of less interest to the general public
- The description of the new annotations could be improved, and authors could release tools to check if their annotations match the features that readers would have downloaded from other sources, to make sure they are consistent. This is an important aspect for practical reproducibility of the work.

**Justification For Why Not Higher Score:**

The paper makes a number of significant contributions (a new feature extraction pipeline, releasing new annotations, providing several measurements and ablations that demonstrate good performance), but is valuable mostly to the deaf and hard of hearing community (please also use this term consistently in the paper, rather than just "deaf"). (1) The new annotation pipeline (as well as the released annotations) are very specific to the sign language recognition task, so it may be best to target this paper to a corresponding conference (Computer Vision, ACM Multimedia, or HCI, etc). ICLR audience would be more appropriate if the methods could be shown to generalize to other similar tasks. (2) the description of the new annotations could be improved, and authors could release tools to check if their annotations match the features that readers would have downloaded from other sources, to make sure they are consistent. (3) the paper does provide a large number of numbers and ablations, but the AC shares one of the reviewers' concerns that these add a lot of information, but it is less clear how much generalizable insight e.g. specific FID scores provide - the target audience would probably be a narrow part of ICLR audience.

**Justification For Why Not Lower Score:**

n/a

---

### Decision · Program_Chairs · 2024-01-16

Reject